# Satellite and Ground-based Measurements of $X_{CO_2}$ in a Remote Semi-Arid Region of Australia

Voltaire A. Velazco [1], Nicholas M. Deutscher [1], Isamu Morino [2], Osamu Uchino [2], Beata Bukosa [1], Masataka Ajiro [2], Akihide Kamei [2], Nicholas B. Jones [1], Clare Paton-Walsh [1], and David W. T. Griffith [1]

[1]Centre for Atmospheric Chemistry, School of Earth, Atmospheric and Life Sciences, Faculty of Science, Medicine and Health, University of Wollongong, Australia
[2]Satellite Remote Sensing Section and Satellite Observation Center, Center for Global Environmental Research, National Institute for Environmental Studies (NIES), Onogawa 16-2, Tsukuba, Ibaraki 305-8506, Japan

**Correspondence:** Voltaire A. Velazco (voltaire@uow.edu.au)

**Abstract.** In this study, we present ground-based measurements of column-averaged dry-air mole fractions (DMFs) of $CO_2$ (or $X_{CO_2}$) taken in a semi-arid region of Australia with an EM27/SUN portable spectrometer equipped with an automated clamshell cover. We compared these measurements to space-based $X_{CO_2}$ retrievals from the Greenhouse Gases Observing Satellite (GOSAT). Side-by-side measurements of EM27/SUN with the Total Carbon Column Observing Network (TCCON) instrument at the University of Wollongong were conducted in 2015-2016 to derive an $X_{CO_2}$ scaling factor of 0.9954 relative to TCCON. Although we found a slight drift of .13% over three months in the calibration curve of the EM27/SUN vs TCCON $X_{CO_2}$, the alignment of the EM27/SUN proved stable enough for a two-week campaign, keeping the retrieved $X_{air}$ values, another measure of stability, to within 0.5% and the modulation efficiency to within 2%. From the measurements in Alice Springs, we confirm a small bias of around 2 ppm in the GOSAT M-gain to H-gain $X_{CO_2}$ retrievals, as reported by the NIES GOSAT validation team. Based on the reported random errors from GOSAT, we estimate the required duration of a future campaign in order to better understand the estimated bias between the EM27/SUN and GOSAT. The dataset from the Alice Springs measurements is accessible at http://dx.doi.org/10.4225/48/5b21f16ce69bc (Velazco et al., 2018).

## 1 Introduction

The Greenhouse Gases Observing Satellite (GOSAT), launched on January 23, 2009, is the first satellite in orbit dedicated to the measurements of the two major anthropogenic greenhouse gases that cause global warming: $CO_2$ and $CH_4$ (Yokota et al., 2009; Kuze et al., 2009). Since its launch, GOSAT data have been used to study and determine the global distributions of $CO_2$ and $CH_4$ (Reuter et al., 2014; Houweling et al., 2015; Parker et al., 2015; Turner et al., 2015). The spectrometer on board GOSAT, called Thermal and Near infrared Sensor for Carbon Observation - Fourier Transform Spectrometer (TANSO-FTS), is able to simultaneously detect short wave infrared (SWIR: bands 1 at 0.76 $\mu$m, 2 at 1.6 $\mu$m and 3 at 2.0 $\mu$m) and thermal infrared (TIR: band 4 from 5.5 to 14.3 $\mu$m). The TANSO-FTS has a footprint of about 10.5 km in diameter on the ground viewed at nadir, which is observed for 4 seconds. Bands 1, 2, and 3 are measured in two linear polarizations simultaneously with three selectable gains; high (H), medium (M) and low (L) (Suto et al., 2013). GOSAT employs the different gain settings

to compensate for the different signal levels due to the reflective properties (albedo) of the Earth's surface, which also depend on wavelength. These gain settings are prespecified at certain locations because GOSAT does not observe both M and H gains simultaneously. For the majority of the soundings over land, H-gain is used. GOSAT M-gain retrievals over land are used over surfaces that are bright in the SWIR such as deserts and semi-arid regions. However, a bias between GOSAT M- and H-gain retrievals of $X_{CO_2}$ has been reported, and there has been a lack of M-gain validation with TCCON (Yoshida et al., 2013). In Australia where surface reflectivity values are generally high, GOSAT was configured to observe using M gain for much of the land surface, at first to avoid detector saturation. However, it was found that some observations using H gain did not result in saturated signals and were still useful. Therefore from 2012-Feb-11, GOSAT started to perform alternate observations using both H and M gains in order to investigate the differences from retrievals between these two gain settings.

Fig. 1 shows the locations of GOSAT M-gain soundings over a period of one year obtained from the NIES (National Institute for Environmental Studies, Japan) version 2.72 $X_{CO_2}$ retrieval algorithm (white squares) and data from the previous version 2.60 (black dots) for reference. Satellite-based retrievals of $X_{CO_2}$ are validated by the Total Carbon Column Observing Network (TCCON), a ground-based network dedicated to the precise and accurate measurements of greenhouse gases (Wunch et al., 2011). TCCON data are used to validate satellite instruments such as GOSAT (Morino et al., 2011), GOSAT-2 (Matsunaga et al., 2018), OCO-2 (Crisp et al., 2017) and Sentinel 5P (Borsdorff et al., 2018). However, apart from locations in the western U.S.A. (Dryden and Los Alamos TCCON sites), there are no operational TCCON stations (magenta stars in Fig. 1) that are ideally located in a clean desert or semi-arid region not influenced by nearby anthropogenic emissions and have plenty of nearby GOSAT M-gain soundings. The Dryden TCCON station at the Armstrong Flight Research Center (AFRC), Edwards, CA is located in the Mojave desert at 34.960° N, 117.881° W (700 m a.s.l). Although not as densely populated, AFRC is only approximately 100 km north of Los Angeles (pop. c. 17.8 million) and 100 km east of Bakersfield, CA (pop. c. 376,380).

The world's deserts and semi-arid regions encompass large areas that are mostly undisturbed by recent anthropogenic emissions and are important for understanding the carbon cycle. Recently, the importance of semi-arid regions in the carbon cycle inter-annual variability has been highlighted (Poulter et al., 2014; Ahlström et al., 2015; Trudinger et al., 2016). If the goal is to continuously improve the accuracy of carbon cycle studies, then the accuracy of satellite retrievals needs to be improved as well, because if observations over high albedo are not available or biased, flux estimates would likely be biased as well and would lead to misinterpretation. Therefore, measurements over M-gain regions are needed by the satellite community (Yoshida et al., 2013) and highlights the significance of this study.

Central Australia has a large semi-arid region that is relatively easy to access. There are plenty of uninhabited, vast, homogeneous areas that can accommodate large footprints of satellite-based sensors such as GOSAT, GOSAT-2, OCO-2, and Sentinel 5P. Central Australia is also an ideal place to measure undisturbed atmospheric conditions that can serve as calibration point for satellite retrievals of atmospheric composition. At the same time, the desert environment has high surface reflectivity, which is a challenge for satellite retrievals because aerosols, depending on type, can lead to an effect called optical path lengthening and this effect is dominant in regions with high albedo (Yoshida et al., 2013). Recently, Iwasaki et al. (2019) showed that there is an increase in the $X_{CO_2}$ retrievals using their PPDF-S algorithm when the albedo at 1.6 $\mu$m was high, implying that the retrieved $X_{CO_2}$ is strongly related to the surface albedo. This challenge leads to the improvement of satellite retrievals. Despite

the importance of desert locations like Central Australia in remote sensing, infrastructure support is not available and accessibility is a challenge. Benchmark measurements and pilot studies for desert sites are needed to assess the benefit and feasibility of such sites because setting up a TCCON site in remote deserts will be difficult logistically and financially. We address this need by utilizing a well-established portable solar-viewing spectrometer, an EM27/SUN by Bruker Optics GmbH (Gisi et al., 2012; Frey et al., 2015; Hase et al., 2015; Hedelius et al., 2016; Frey et al., 2019), which was retrofitted with a protective fairing and automated solar tracker clamshell cover for operations in a harsh environment. For brevity, we will refer to this instrument as EM27. The instrument can measure spectra covering the spectral bands in the near infrared necessary to derive column-averaged dry-air mole fractions of $CO_2$, $CH_4$ and CO with sufficient stability for short-term campaigns (Hedelius et al., 2016). The instrument was transported to Alice Springs in Central Australia with the primary objective of validating the GOSAT $X_{CO_2}$ signal and making benchmark measurements in the region. The Alice Springs measurements are unique considering that they have been collected from a clean desert environment where GOSAT M-gain soundings are also abundant and close enough (within 100 km) to compare with the EM27.

This manuscript is organized as follows. A description of Alice Springs and desert Australia is given in Section 2. In Section 3, we briefly discuss the instruments and methods, which are already well established. Results of measurement comparisons with the TCCON station in Wollongong and comparisons with the GOSAT M-gain and H-gain soundings in Alice Springs are shown and discussed under Section 4. We focus on $X_{CO_2}$ measurements from the EM27 and GOSAT. We compare NIES GOSAT retrieval versions 2.60 and the new version 2.72, which are both not bias corrected, with the retrievals from the EM27. Statistical calculations and recommendations for a future Alice Springs campaign are also under Section 4. We provide our conclusions in Section 5.

## 2   Alice Springs Australia Site Description

We conducted a measurement campaign with the EM27 system at the Australian Bureau of Meteorology (BOM) facility in Alice Springs (23.79°S, 133.89°E) from September 29 to October 6, 2016, with the primary objective of validating the GOSAT $X_{CO_2}$ signal in this semi-arid region. Alice Springs is located in Central Australia, also called the "Red Centre". Next to Darwin and Palmerston, it is the third-largest town in the Northern Territory of Australia with a population of 23,726 (2016 census). The vegetation around Alice Springs is composed mostly of dry scrubby grassland. The Alice Springs terrain, consisting mostly of sandy plains with some areas of rocky highland, is bounded by several deserts; the Tanami desert to the north, Simpson desert to the east and southeast, the Great Victoria desert to the south and the Gibson desert to the west. In September, statistical data from the Australian Bureau of Meteorology (BOM) show an average monthly rainfall of 8.7 mm, average daily high and low temperatures of 27.3 °C and 10.3 °C respectively and mean monthly sunshine of 300 hours. Collected GOSAT M-gain soundings over land for one whole year shown in Fig. 1 include a large part of this "Red Centre" including Alice Springs. Note that for Ver. 02.72 FTS SWIR L2 retrievals, upgraded input and reference products were used. For example, there was an improvement in the spatial resolution of the cloud flagging procedure, which employs the CAI L2 (Cloud and Aerosol Imager, Level 2) data. This improvement resulted in better screening of the data and may have resulted in an increased number of

soundings that were passed for the Ver. 02.72 FTS SWIR L2 retrievals. Apart from remote regions in North Africa, the Middle East and near-densely-populated areas in California USA, Central Australia is the only region that provides an abundant amount of M-gain soundings.

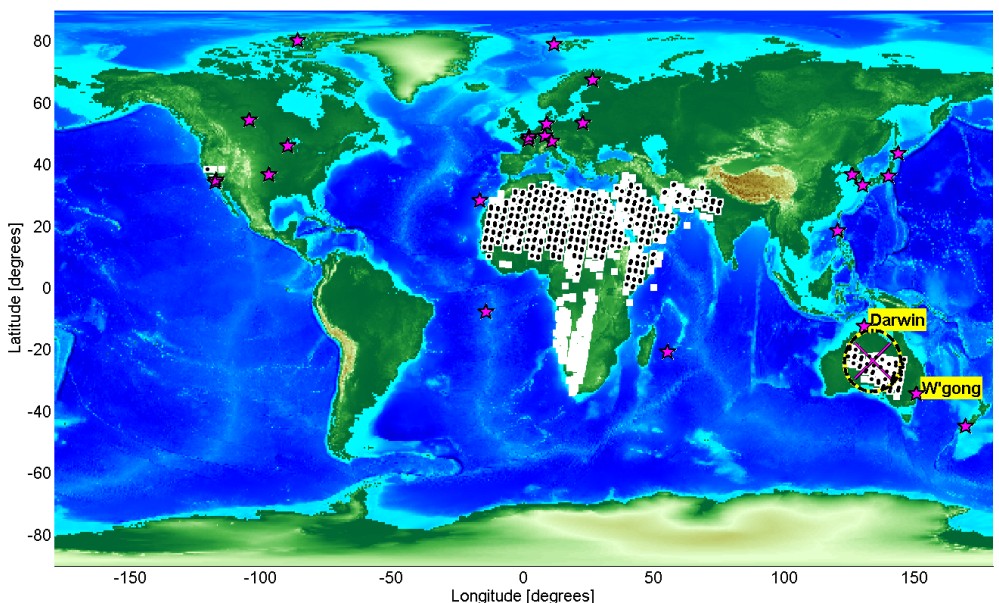

**Figure 1.** Locations of GOSAT M-gain soundings for 2016. Black dots are from version 2.60 and the white squares are from version 2.72. Magenta stars indicate the location of operational TCCON sites as of October 2018. The magenta cross marks the location of Alice Springs in Australia and the magenta circle has a radius of approximately 1000 km centered on Alice Springs. The Darwin and Wollongong (W'gong) sites in Australia are labeled.

## 3 Instruments and Methods

### 3.1 EM27/SUN with Automated Clamshell Cover

The EM27 system and its characterization are thoroughly described in the works of Gisi et al. (2012); Frey et al. (2015); Hase et al. (2015) and Hedelius et al. (2016). For total column measurements of $CO_2$, $CH_4$, $H_2O$ and $O_2$ spectra in the near infrared, the instrument is fitted with an Indium Gallium Arsenide (InGaAs) detector dedicated to 5,000–12,000 $cm^{-1}$. This also enables measurements of spectra covering the $O_2$ bands necessary to derive column-averaged dry-air mole fractions of $CO_2$ and $CH_4$ similar to the method used by TCCON. TCCON uses a maximum optical path difference (MOPD) of 45 cm, corresponding to a spectral resolution of 0.02 $cm^{-1}$. However, in contrast to TCCON, the EM27 we used has the typical MOPD of 1.8 cm, corresponding to a spectral resolution of 0.5 $cm^{-1}$.

We equipped the Bruker EM27 with a weather station-controlled automated clamshell cover for the solar tracker and protective fairing (Fig. 2). Our objective was to achieve autonomous and remote measurements of greenhouse gases in harsh environments, in particular, to perform much needed measurements of $X_{CO_2}$ and $X_{CH_4}$, in desert Australia. The design and more details of this construction can be obtained by contacting the authors.

Recently, Hedelius et al. (2016) published a long-term assessment of errors and biases in retrievals of several TCCON gases from the EM27. In spite of a reported drift, they found that the stability of the EM27 is sufficient for short-term campaigns. Therefore, before deployment to Alice Springs, we have operated the EM27 from December 2015 to September 2016 and tested the automated clamshell cover at the University of Wollongong, where the TCCON instrument (Griffith et al., 2014) is also located. Results of the collocated measurements are in Section 4.

## 3.2 Retrieval of $X_{CO_2}$ and $X_{air}$

To retrieve $X_{CO_2}$, we use the same retrieval software used by TCCON called GGG2014 (Wunch et al., 2017), which was also used by Hedelius et al. (2016) for their EM27. Column-averaged dry-air mole fractions (DMFs) of gases ($X_{gas}$) are retrieved from the EM27 measurement as in Wunch et al. (2010):

$$X_{gas} = \frac{VC_{gas}}{VC_{dryair}} = 0.2095 * \frac{VC_{gas}}{VC_{O_2}} \tag{1}$$

where $VC_{gas}$ is the vertical column of the gas. $VC_{dryair}$ is the dry pressure column of air and 0.2095 is the known DMF of oxygen. The DMF of air ("$X_{air}$") can be calculated using the measured oxygen column from the EM27 spectrum:

$$X_{air} = \frac{VC_{air}}{VC_{O_2}} * 0.2095 - X_{H_2O} * \frac{mH_2O}{m_{air}^{dry}} \tag{2}$$

where $mH_2O$ and $m_{air}^{dry}$ are the mean molecular masses of water (18.02 g mol$^{-1}$) and dry air (28.964 g mol$^{-1}$), $X_{H_2O}$ is the retrieved DMF of water vapour and $VC_{air}$ is calculated from the surface pressure, $P_s$:

$$VC_{air} = \frac{P_s}{g * \frac{m_{air}^{dry}}{N_a}} \tag{3}$$

where $g$ is the column-averaged acceleration due to gravity and $N_a$ is Avogadro's constant. $X_{air}$ is a good indicator of instrument stability and changes in spectrometer alignment because 1) $VC_{air}$ is calculated using the surface pressure, which is independently measured by a pressure sensor to better than 0.3 hPa, keeping accuracy over long periods (Wunch et al., 2011); and 2) the atmospheric oxygen column is not particularly variable in dry air, hence the retrieved $VC_{O_2}$ by the spectrometer should be close to constant. From Eq. (2) it follows that a perfectly accurate measurement would lead to an $X_{air}$ value of unity, however due to inaccuracies in the $O_2$ spectroscopy, the actual value is approximately 0.98 for all TCCON sites. We also find that $X_{air}$ values well outside of the $2^{nd}$ and $98^{th}$ percentiles may indicate an obstruction in the solar beam (e.g. birds, fast moving clouds, leaves from trees, etc.). Therefore the $X_{air}$ can also serve as data filtering criteria.

In Wollongong, both the TCCON and EM27 solar tracker covers were opened and closed by the same pneumatic system. The same weather station provided the meteorological data that were used to pre-filter the data (e.g. fractional variation in solar

intensity, wind speed and direction, pressure, etc). For this study, we did not filter the data according to solar zenith angles (SZAs) anymore, instead in addition to the pre-filter, we filtered out noisy retrievals by selecting only those with $X_{air}$ values within 0.5 and 1.5 because anything beyond that would be unrealistic in the atmosphere but most likely be the cause of an interference or obstruction.

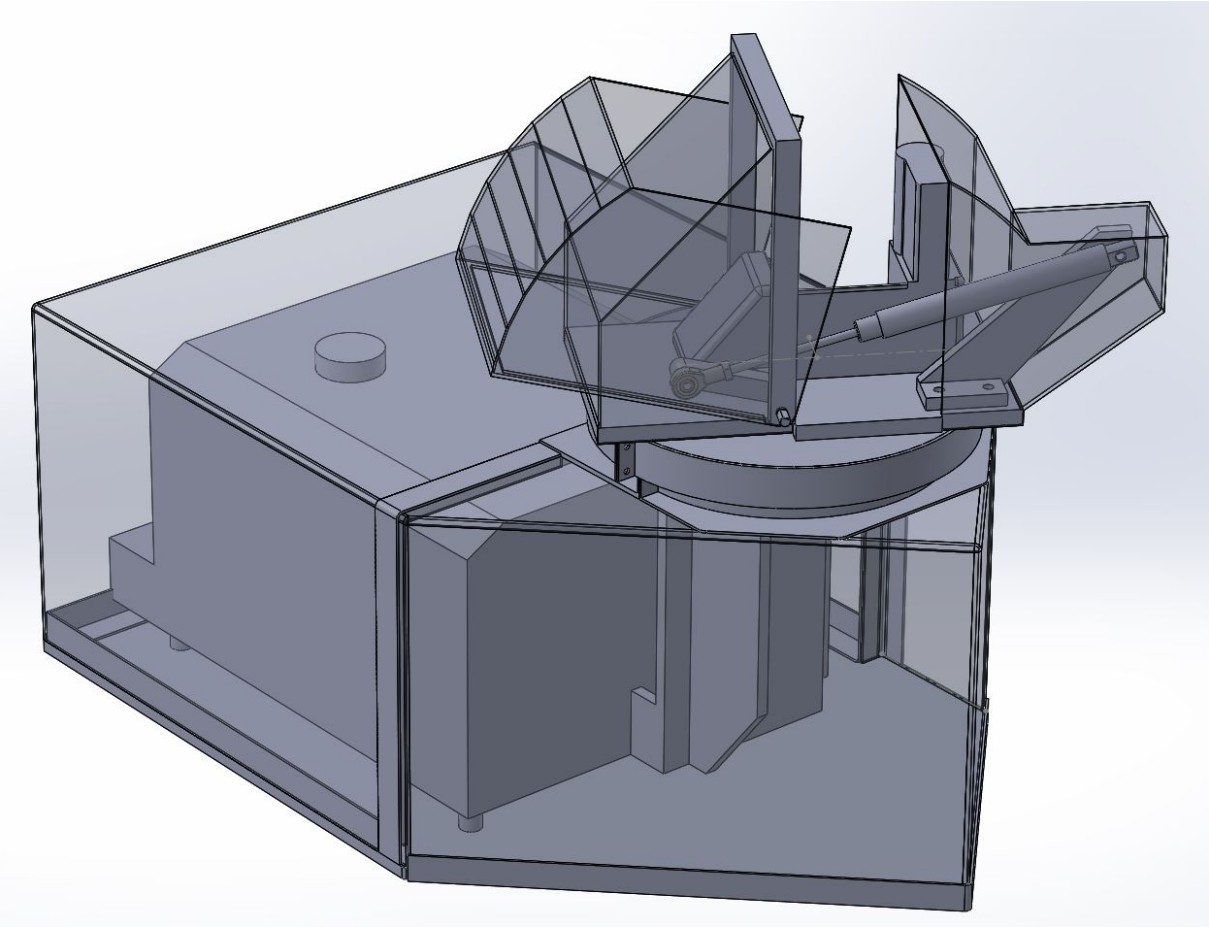

**Figure 2.** Drawing of the Wollongong EM27 automated clamshell cover and fairing (by Steve Selby, UOW).

## 3.3 GOSAT Specific Point Observations

The TANSO-FTS on GOSAT has a two-axis pointing system. Normally, TANSO-FTS follows an M shaped grid on a 5-point cross-track scan mode (https://www.eorc.jaxa.jp/GOSAT/instrument_1.html). By using this pointing system to vary the observation geometry, it is able to observe specific points, i.e. it can view targets with angles up to $\pm20°$ along the satellite track and by $\pm30°$ across the track. Specific point observations over Alice Springs were requested from July 2016, in preparation for the campaign in September. Five locations within 100 km of the center of Alice Springs were targeted (see Fig. 3). We only used the specific point observation data to compare with the EM27. However, to construct the time-series shown in Fig. 8, we

used all available GOSAT data version V2.72 from NIES spanning the years 2010 to 2017. Daily averages within 1000 km of Alice Springs are calculated for M- and H-gain retrievals separately.

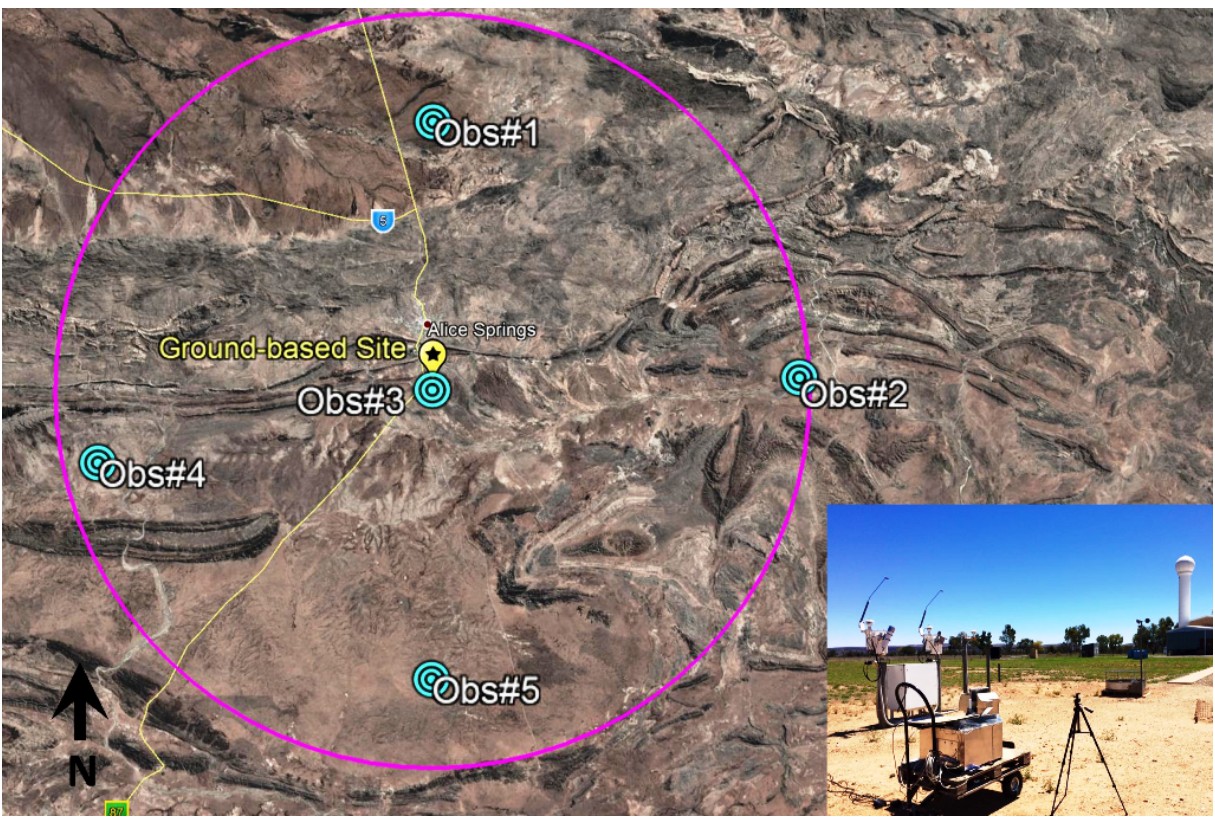

**Figure 3.** A Google Earth map showing the GOSAT specific point observations around Alice Springs (cyan circles labeled "Obs#") . The ground-based site is securely fenced and located at the airport, c. 14 km south of the town center. The magenta circle shows a 60-km radius from the site, for visual reference. The inset shows the EM27 during the measurements.

## 4 Results

### 4.1 Comparison with Wollongong TCCON Station

5   In this subsection, we present a side-by-side comparison of retrievals of $X_{CO_2}$ between the EM27 and the Wollongong TCCON station. Apart from the works by Hedelius et al. (2016) and Frey et al. (2019), studies on long-term comparisons of the EM27 with TCCON are rare. Here, we focus on comparisons of $X_{air}$ and $X_{CO_2}$ from measurements spanning almost one year (Nov. 2015 to Sep. 2016) under varied environmental conditions. Both instruments normally measure at the same time, apart from interruptions due to occasional mid-infrared measurements with the 125HR or rare software glitches (e.g. JAVA issues).

### 4.1.1 $\mathbf{X_{air}}$

As mentioned in Sect. 3.2, inaccuracies in the $O_2$ spectroscopy lead to an $X_{air}$ value of approximately 0.98 for all TCCON sites. This value varies by approximately 1% with solar zenith angle, indicating an airmass dependence in the $O_2$ retrievals (Pollard et al., 2017). Deviations from the characteristic values for $X_{air}$ are generally indicative of erroneous behavior in the measurement and retrieval system such as interferometer misalignment, tracking errors or fitting to an incorrect airmass due to timing errors, as recently reported in Pollard et al. (2017).

Figure 4 shows the normalized probability distribution functions (PDF) of $X_{air}$ values retrieved by both instruments in 2016. One manifestation of instrumental differences between the Wollongong TCCON and the EM27 is the difference in the $X_{air}$ retrieved from both instruments. The corresponding mean values of $X_{air}$ from both instruments are 0.9824 and 0.9849 for TCCON and EM27 respectively. The TCCON $X_{air}$ PDF exhibits a more Gaussian pattern, while the EM27 $X_{air}$ PDF is slightly skewed.

Figure 5 shows the mean $X_{air}$ values calculated for each month in 2016 for TCCON (red) and EM27 (blue). The error bars represent the spread (one standard deviation) in the monthly mean values of $X_{air}$. This gives us a measure of how much the $X_{air}$ changes and therefore another measure of instrument stability. In Fig. 5, we see that the mean $X_{air}$ values from the EM27 are slightly larger compared to the TCCON instrument but a slight seasonal dependence can be seen in both instruments. The difference in the mean $X_{air}$ values does not appear to be purely related to sampling differently across that seasonal dependence. The SZAs seem to have an effect as well, although small. To show this in Fig. 5, we filtered $X_{air}$ values for SZA $\leq 45\,^\circ$ (black dots and circles) and above $45\,^\circ$ (grey pluses and crosses). The SZA effect is within approximately 1% for both instruments. We note that there were no re-alignments done on the EM27 and TCCON instruments during this period, however the EM27 clamshell cover and fairing were fitted on the EM27 in February 2016. It is possible that this may have affected the alignment resulting in a shift in the $X_{air}$.

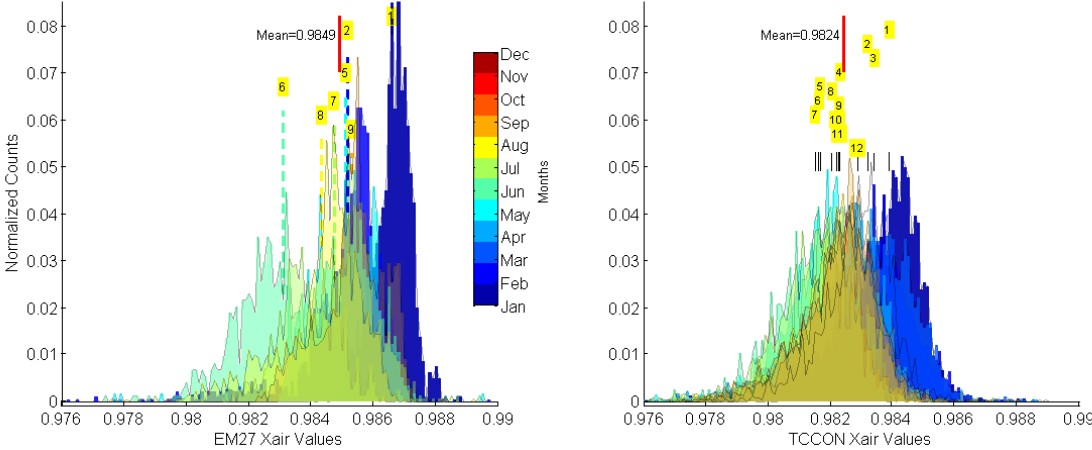

**Figure 4.** Normalized probability distribution functions (PDF) of $X_{air}$ from TCCON (right panel) and $X_{air}$ from the EM27 (left panel) in Wollongong for 2016. The PDFs for each month are separated by colors and the positions of the mean $X_{air}$ monthly values are indicated by the month number. The mean $X_{air}$ for the whole year for the EM27 is 0.9849 and 0.9824 for TCCON. The clamshell cover and fairing were installed on the EM27 in February, this may have affected the alignment resulting in a shift in the $X_{air}$.

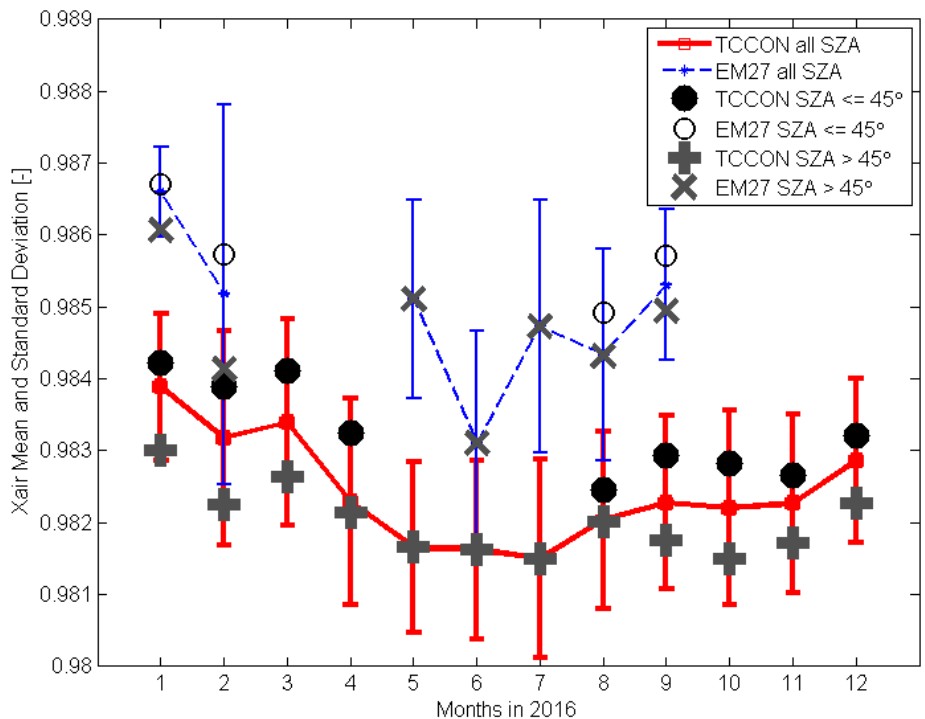

**Figure 5.** A measure of the spread of the of $X_{air}$ values from TCCON (red) and EM27 (blue) by taking the mean (central markers) and corresponding standard deviations (bars) for each of the monthly probability distribution functions shown in Fig. 4. Filtered $X_{air}$ values for angles $\leq 45\,°$ (black dots and circles) and above $45\,°$ (grey pluses and crosses) are also shown for reference.

### 4.1.2  $X_{CO_2}$ Comparison with Wollongong TCCON and Apparent Drift

From the EM27 and TCCON measurements in 2016, we derived a scaling factor for the EM27. We scaled the EM27 measurements to Wollongong TCCON data because the Wollongong $X_{CO_2}$ has been calibrated against aircraft profile measurements that are traceable to the WMO in-situ scale (Wunch et al., 2010). The retrieval method is predicted to be both linear and have zero intercept (Wunch et al., 2010). Therefore, we fit hourly mean data from TCCON and EM27 employing linear least squares and force a zero intercept. The standard errors of the weighted means are used as weights in the fit. From this exercise, we arrive at a scaling factor of EM27=0.9954*TCCON (Fig. 6). Frey et al. (2015) reported a comparable scaling factor of 0.9951*TCCON for $X_{CO_2}$ using collocated measurements of five EM27 instruments with the TCCON instrument at the Karlsruhe Institute of Technology in Germany. However, Hedelius et al. (2016) reported a smaller bias of +0.03% between EM27 and TCCON. We think that the slightly larger bias in this work (-0.46%) is probably due to an imperfect alignment of our EM27, with a modulation efficiency (ME) of 96% at maximum optical path difference (OPD). The ME calculation was done using water lines measured at ambient laboratory/room air (Frey et al., 2015). The ILS was monitored before, during and

after the campaign and the ME remained stable within 94%-96%. Therefore, we are confident that the scaling factor relative to TCCON was consistent during the campaign.

Hedelius et al. (2016) reported a noticeable drift in the EM27 measurements over several months. We also observed a very small drift in the EM27 vs TCCON $X_{CO_2}$ values shown in Fig. 6, where the calibration line has been derived. For clarity, we color-coded the EM27 vs TCCON $X_{CO_2}$ values in Fig. 6 according to days after Jan 1, 2016. The colors progress with time, i.e. from dark blue (oldest) to dark brown (most recent). We found that the drift within three months before the campaign is very small, i.e. 0.13% over three months as calculated from the EM27/TCCON scaling factors from July to mid September. We did not consider the averaging kernels in this work. The averaging kernels of the EM27 have been previously presented and compared to TCCON in a study by Hedelius et al. (2016). In their study, they found that although there are differences in the TCCON and EM27 averaging kernels, the effect of the differences in averaging kernels from the top of the atmosphere cancel out the effect of differences at the bottom. Further work and more measurements may be necessary to better understand the cause of this phenomenon but this is beyond the scope of this study.

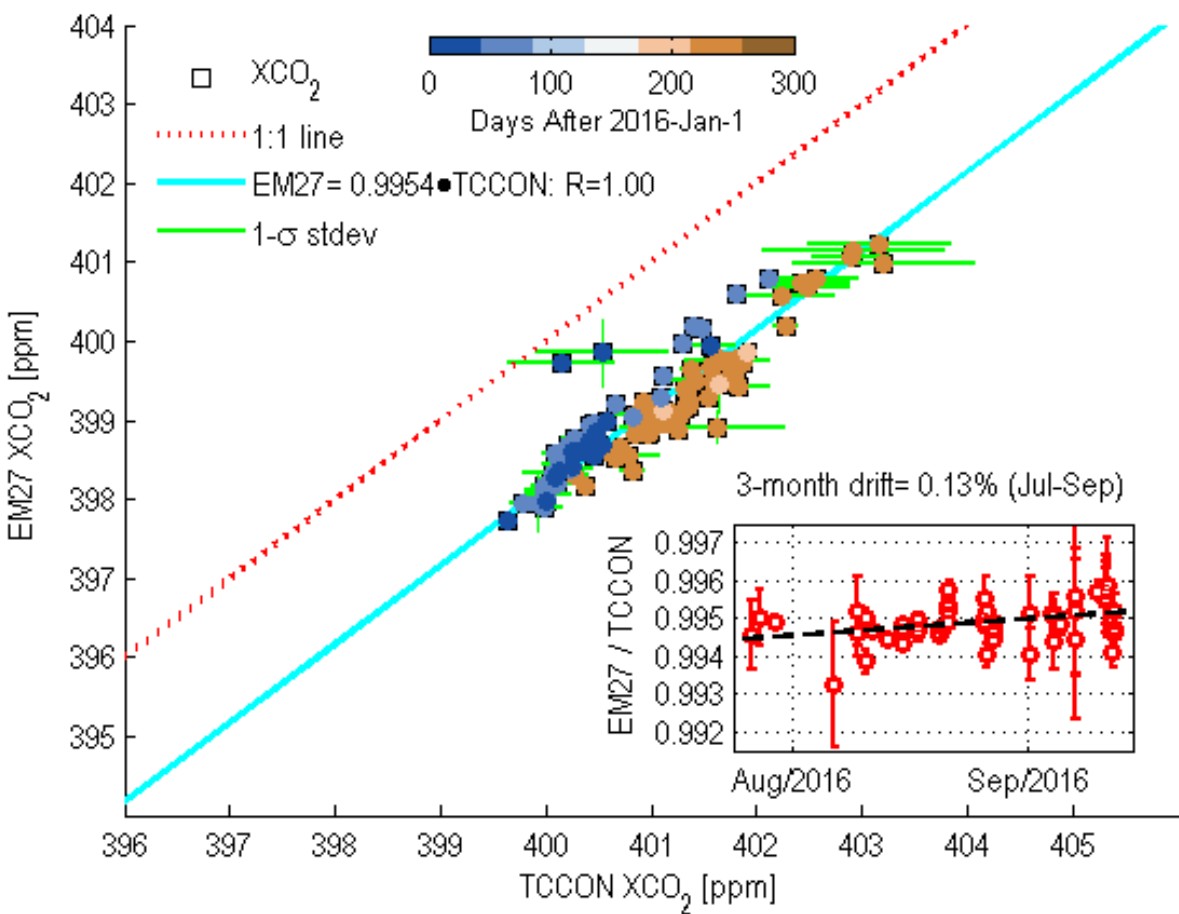

**Figure 6.** Weighted hourly mean $X_{CO_2}$ from TCCON vs $X_{CO_2}$ from EM27 in Wollongong for 2015-2016. To avoid noisy data, only measurements corresponding to TCCON with retrieved $X_{air}$ values within 0.9783 and 0.9853 were selected ( $X_{air}$ values within the $2^{nd}$ to $98^{th}$ percentile). The inset shows the drift in the hourly weighted mean scaling factors over three months (red dots). The error bars are the sum, in quadrature, of the relative standard deviations of the weighted hourly means from each instrument.

## 4.2 Measurements in Alice Springs

Results of the the ground based $X_{CO_2}$ measurements during the Alice Springs campaign are shown in Fig. 7. The measurements started on September 29, 2016 with interruptions due to cloudy and/or rainy weather. In total, we gathered 6 days of good ground-based measurements of $X_{CO_2}$ under good weather conditions, which coincided with 9 GOSAT specific point soundings

5   within 100 km of the site. A slight airmass dependence in the $X_{CO_2}$ retrievals results in lower $X_{CO_2}$ values at low sun elevation, as can be observed in Fig. 7. This airmass dependence is well-known and also discussed in Wunch et al. (2011). But for the purpose of GOSAT comparisons, this airmass dependence is negligible because GOSAT passes the site at around 13:05 local time (03:35 UTC), corresponding to a high sun elevation (see Table 1 for the dates).

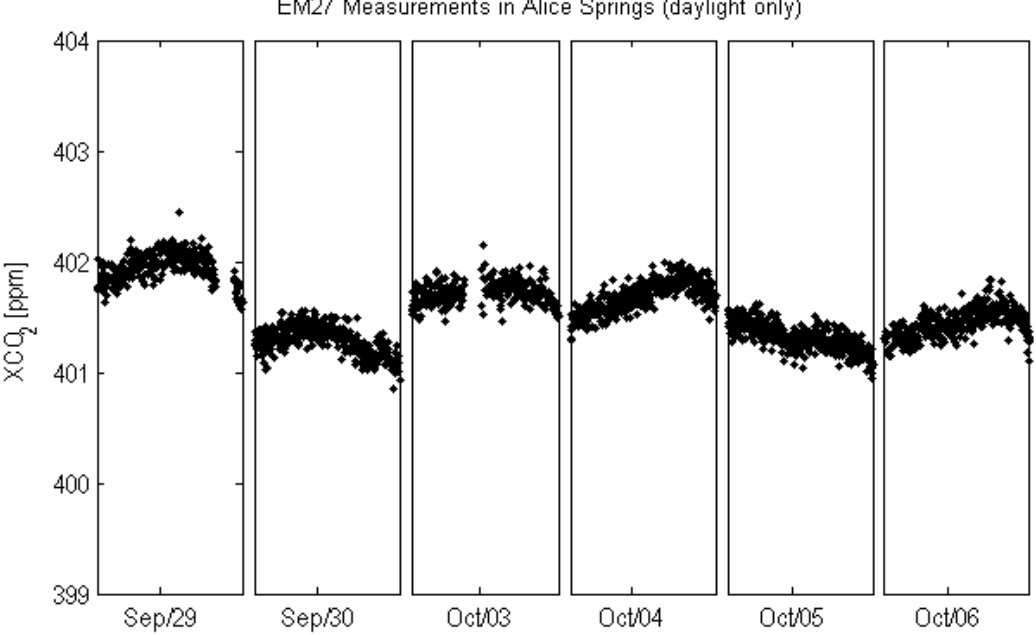

**Figure 7.** Retrievals of $X_{CO_2}$ from the EM27 in Alice Springs, compressed along the x-axis (days). The measurement days on the x-axis are not continuous due to interruptions from bad weather. See Sec. 6 for data availability.

### 4.3 Comparisons with GOSAT Measurements in Alice Springs

To demonstrate the variability of $X_{CO_2}$ in the region, we plot in Fig. 8 the daily mean time series of GOSAT soundings using these two coincidence criteria: 1) all soundings within a 1000-km radius centered at the BOM facility in Alice Springs and 2) all soundings measured within the same day (local time). Each data point considered in the calculation of the daily mean is

weighted by the corresponding reported retrieval error for that particular GOSAT sounding. Vertical gray lines represent the corresponding standard deviation for each daily mean calculation. The H-gain retrievals (red triangles) and M-gain retrievals (blue squares) are separated in this plot. Weighted linear least squares fitted straight lines were calculated for the M-gain $X_{CO_2}$ (cyan) and H-gain $X_{CO_2}$ data (magenta). We confirm that the M-gain retrievals are biased high (around 2 ppm) compared to the H-gain retrievals. Nevertheless, both least squares fitted lines to the H- and M-gain retrievals show an increase of about

2.28 ppm/year with y-intercept values at 133.0 ppm (H-gain) and 134.92 ppm (M-gain).

Compared to the TCCON sites in Wollongong and Darwin (Deutscher et al., 2010, 2014), the $X_{CO_2}$ signal in Alice Springs is relatively smooth and undisturbed, which is expected because Alice Springs is in the middle of the Australian continent, a desert environment with no large sources to interfere with the $X_{CO_2}$ signal. The terrestrial biosphere is the largest driver of variability in the Southern Hemisphere column $X_{CO_2}$ (Deutscher et al., 2014). Moreover, Deutscher et al. (2014) have shown

that the magnitude of the seasonal variability in the column-average dry-air mole fraction of $X_{CO_2}$ is comparable in magnitude

to the annual increase. In Alice Springs, only a slight seasonal cycle in the measured $X_{CO_2}$ can be seen. This slight cycle is probably less driven by the surrounding vegetation because the centre of the Australian arid zone is not affected by seasonality and has low aggregate vegetation (Lawley et al., 2011) but more by meridional effect.

Rainfall records indicate that April-August 2011 had the least amount of rainfall on record in Alice Springs after 2002, which was only broken by 2015, then by 2017 (http://www.bom.gov.au/climate/data/, station number 015590). This significant absence of rain in the region could result in stunted vegetation growth or mortality, which could have led to brighter surfaces. Bright surfaces mean more M-gain measurements and this may explain the 74% increase in M-gain soundings in April-August 2011 compared to the April-August 2010-2017 average of c. 220 soundings/month. In contrast, the number of H-gain retrievals seems to have diminished around 2011, coinciding with the dry months starting from April 2011. The number of H-gain soundings from April-August 2011 was 23% fewer compared to the April-August average from the years 2010-2017 (c. 450 soundings/month).

GOSAT specific point observations that were scheduled during the campaign are shown on Table 1. The times correspond to the times when the satellite is directly above the site. The satellite normally performs 5 observation points across track, with an interferometric scan time lasting 4 seconds (Shiomi et al., 2006). But specific point observations deviate from this pattern by pointing and maximizing observations near the target. M- and H-gain retrievals (6 days apart) were obtained during the short campaign. We averaged and compared the GOSAT retrievals directly with the coincident measurements from the EM27 ($\pm$ 0.5 hour) and as in Sec. 4.1.2, we fit the data using weighted linear least squares and force a zero intercept. From here, we derived a scaling factor of 0.9927*EM27 for GOSAT H-gain and 0.9983*EM27 for the GOSAT M-gain, as shown in Fig. 9. This is only slightly different from the previous version 2.60 of the data (H-gain: 0.9935*EM27 and M-gain: 0.9997*EM27). Indeed, there is a slight bias in the M- and H-gain retrievals from GOSAT, i.e. M-gain retrievals are biased slightly low and H-gain retrievals are very close compared to the EM27. However, the M- and H-gain bias relative to each other has been improved in the version 2.72 data release.

**Table 1.** GOSAT Specific Point Observation Opportunities Over Alice Springs

| Date (UTC) | Orbit Number | Gain Type (H or M) | Remarks |
|---|---|---|---|
| 2016/09/27 03:35 | 4 | - | soundings > 100 km |
| 2016/09/30 03:35 | 4 | H | 4 soundings |
| 2016/10/03 03:35 | 4 | - | soundings > 100 km |
| 2016/10/06 03:35 | 4 | M | 5 soundings |

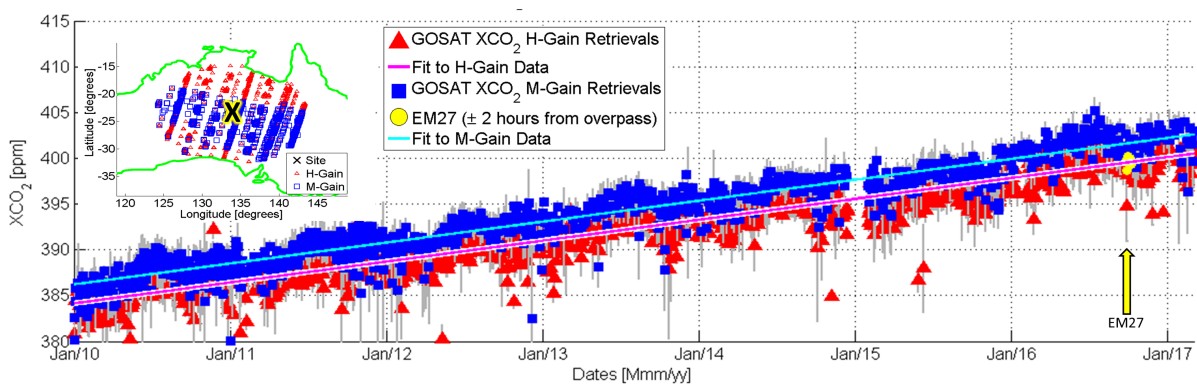

**Figure 8.** Time series $X_{CO_2}$ from GOSAT M-gain and H-gain retrievals in Alice Springs from the NIES version 2.72 product. The timing of the Alice Springs campaign from Sep. 27 to Oct. 6 is indicated by the yellow arrow. Gray lines represent one standard error associated with each daily mean. The inset shows a map projection with the approximate locations of the soundings used for this plot. Only data within a 1000-km radius from Alice Springs (black X on the map) were considered.

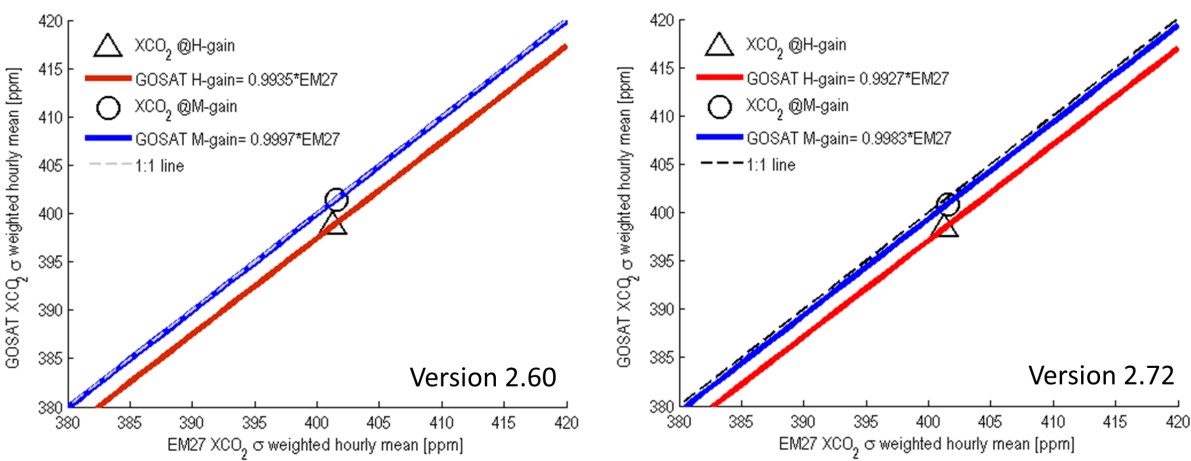

**Figure 9.** Mean $X_{CO_2}$ from the EM27 vs GOSAT $X_{CO_2}$ version 2.60 (left panel) and version 2.72 (right panel) in Alice Springs. The calculated means are weighted by the 1-$\sigma$ uncertainty of the individual measurements. The scaling factor derived from the TCCON comparisons, EM27=0.9954*TCCON, has been taken into account.

## 4.4 Recommendations for a Future Alice Springs Campaign: Simulating Error Reduction in Bias Estimates

In this subsection, we attempt to estimate how long a measurement campaign in Alice Springs should last in order to improve the statistical errors in the estimated bias between GOSAT and the EM27. In order to estimate the required measurements from the EM27, we make the following assumptions and conditions:

5   1. At least one specific point overpass per week (M-gain or H-gain), similar to this campaign.

2. The standard errors of the weighted hourly mean GOSAT specific point observations are: 0.25 ppm and 0.26 ppm for M-gains (versions 2.6 and 2.72, respectively) and 0.29 ppm and 0.28 ppm for H-Gains (versions 2.6 and 2.72, respectively). These values are taken from the averaged standard errors of the weighted hourly means from all specific point observation data falling within 100 km from the site for 2 Sep. 2015 - Feb. 2017.

3. 1-$\sigma$ standard deviation of all EM27 measurements is 0.05 ppm at an averaging time of 1-hour. Chen et al. (2016) reported a 1-$\sigma$ standard deviation of 0.04 ppm at 10 minutes averaging.

4. The error in the estimated bias is calculated as $\sigma_T = \sqrt{\sigma_{GOSAT}^2 + \sigma_{EM27}^2}$, which then improves by the number of weeks (N), i.e. $\sqrt{\sigma_T^2/N}$

5. Any long-term drifts in the satellite and ground-based measurements are negligible over the whole duration of the campaign.

It is important to keep the statistical errors in the estimated bias as low as possible. Miller (2007) showed that a comparison of surface $CO_2$ concentration data and $X_{CO_2}$ data flux inversions clearly reveals a land-ocean bias in the $X_{CO_2}$ retrievals, even when the bias is only 0.1 ppm. Fig. 10 shows a plot of how the statistical errors in the estimated bias improve with the number of weeks in the campaign. The inset shows the normalized histogram of the reported GOSAT single sounding errors within 100 km of the site. Note that although H-gain retrievals have smaller errors on average, the amount of M-gain retrievals improves the standard error of the weighted mean. A two-week campaign would already reduce the statistical error between GOSAT M-gain and EM27 measurements to less than the GOSAT M-Gain weighted mean standard error (0.26 ppm, v2.72). To achieve a statistical error of 0.1 ppm between GOSAT M-gain and EM27 measurements, it would take about 6 to 7 weeks of measurements or more. On the other hand, around 8.5 weeks for H-gain is needed because the standard error of the hourly weighted mean H-gain retrievals are slightly higher due to fewer soundings near the site. The ideal time window to perform this would be from March to November (Autumn-Winter-Spring). It is best to avoid the summer months (Dec-Feb) because of the desert heat, which could be physically demanding for the operator/s and may affect the operation of an EM27 without a temperature-regulated enclosure. A temperature-regulated enclosure would improve stability and make the EM27 suitable for summer operations even in the high heat (>40°C) of Alice Springs. Note however that our error estimates are conservative and this implies that there are no drifts in the M- and H-gain measurements relative to one another that would require long-term validation.

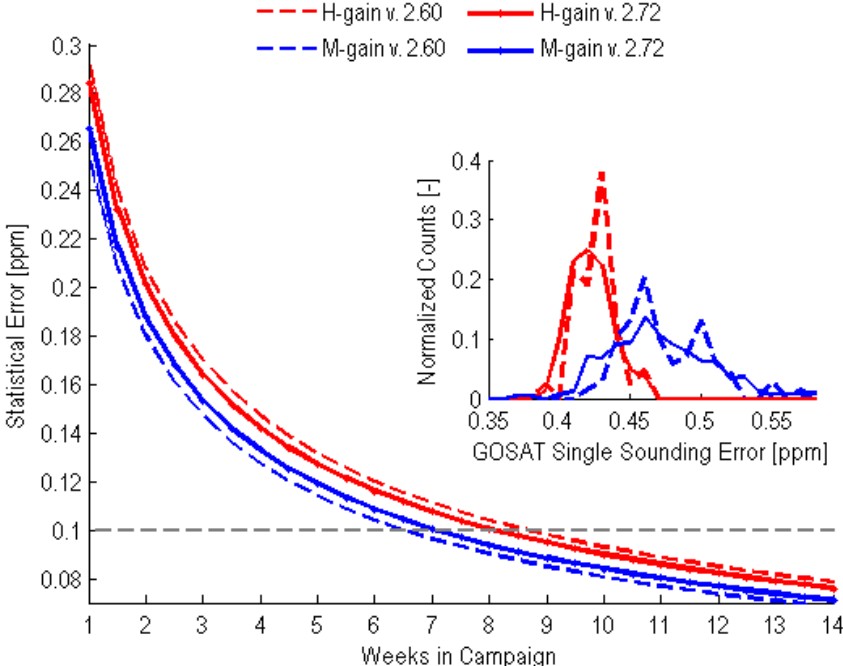

**Figure 10.** The statistical errors in the estimated bias of $X_{CO_2}$ improve with the number of weeks in the campaign. Inset: normalized histogram of the reported GOSAT single sounding errors within 100 km of the site.

## 5  Conclusions

Alice Springs would be a welcome addition to ground sites dedicated to environmental satellite sensor calibration and valida-
tion. With the exception of Lauder, New Zealand, there are no permanently dedicated ground targets for satellite calibration
and validation in the Southern Hemisphere that are far enough from water (to avoid a possible land-ocean bias in retrievals)
and surrounded by homogeneous topography (to avoid a possible altitude bias). Currently, vicarious radiometric calibration of
passive optical instruments from satellites is done on the large homogeneous Railroad Valley playa (RRV) in Nevada, USA.
However plans for future lithium mining threaten to disrupt the playa floor, which could render this critical asset useless for
large footprint sensors on satellites like GOSAT, GOSAT-2, OCO-2 and OCO-3. Alice Springs would offer unique opportu-
nities particularly in the context of satellite surface products and associated emissivity assessments because the environment
of Alice Springs has ideal conditions to measure undisturbed atmospheric conditions that can serve as calibration point for
satellite retrievals of atmospheric composition. At the same time, the desert environment, like RRV, provides high surface re-
flectivity, which is a challenge for satellite retrievals and thus observations are especially needed here for validation purposes.
Subject to funding and interest from the community, a repeat of the campaign is possible. In a future study, we will investigate
GOSAT methane retrievals using the same approach employed here.

## 6 Data Availability

Column averaged dry-air mole fractions of $CO_2$ in Alice Springs, Australia is hosted by the University of Wollongong and can be downloaded via http://ro.uow.edu.au/data/58/ with the DOI: http://dx.doi.org/10.4225/48/5b21f16ce69bc (Velazco et al., 2018). GOSAT data can be obtained from the GOSAT Data Archive Service (GDAS) after registration for access via: https:
5  //data2.gosat.nies.go.jp/index_en.html.

*Author contributions.*

V.V. conceptualized and wrote the manuscript, did the calculations and statistics, made the figures, performed EM27 measurements in Alice Springs, led the EM27 automated clamshell cover project, characterized the EM27 and TCCON ILS and oversaw TCCON measurements in Wollongong during this study.

10  N.D. helped in writing the manuscript, organized and co-acquired funding for the Alice Springs campaign, liaised with BOM, performed the GGG2014 retrievals of the EM27 data, performed measurements in Alice Springs, oversaw the EM27 measurements in Wollongong.

I.M. helped in writing the manuscript, provided ideas, conceptualized the need for statistical calculations (section 4.4).

O.U. presented and highlighted the M- and H-gain bias and the need for validating GOSAT in M-gain region as representative
of the GOSAT project in the Research Announcement (RA) meeting, helped in writing the manuscript and provided advice.

D.G. supervised the EM27 automated clamshell cover project, supervised the EM27 and TCCON measurements, created the preprocessing software, helped in writing the manuscript and provided advise.

N.J. carried out EM27 measurements in Wollongong and helped in determining the EM27 ILS.

C.P.W. performed TCCON measurements in Wollongong and helped improve the manuscript.

B.B. carried out TCCON measurements in Wollongong and helped improve the manuscript.

A.K. and M.A. enabled the specific observation mode in Alice Springs from 2016.

*Competing interests.*  The authors declare that they have no conflict of interest.

*Acknowledgements.*  V.V. would like to thank University of Wollongong SMAH for the advancement grants scheme that funded the fabrication of the EM27 clamshell cover and fairing (Grant number 2016/SPGA-S/07). We would like to thank the Australian Bureau of Meteorology
(BOM), especially site manager Victoria McLean for the support in Alice Springs. We thank Graham Kettlewell for technical advise, programming the weather station and liaising with the workshop. We thank Martin Riggenbach for the design and creation of the weather station interface and for providing technical advise. We thank Steve Selby for the design and fabrication of the EM27 clamshell and cover and Phd student Neil Page for assistance with the EM27 measurements. The specific point observation modes have been made possible by the GOSAT Project Office. Part of this activity has been conducted under the framework of the GOSAT Research Announcement 8 (RA8) Project. N.D.

acknowledges funding from the Australian Research Council (ARC) via DECRA grant DE140100178. Wollongong TCCON activities have been funded through ARC Discovery Projects (DP0879468, DP110103118, DP140101552, DP160101598) and LIEF infrastructure grant LE0668470.

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
