# Peer review of "Satellite and Ground-based Measurements of $X_{\rm CO_2}$ in a Remote Semi-Arid Region of Australia"

_Earth System Science Data, 2018_

## Referee Comment (RC1) · Anonymous Referee #1 · 4 Feb 2019

This study presents ground-based remote sensing measurements obtained in remote central Australia. The primary goal of these ground measurements is to compare with satellite measurements (esp. GOSAT) since there is a paucity of ground-based observations not only in central Australia, but also in other locations worldwide with high albedo surfaces. For ground observations to be useful for satellite comparisons, they need to have a higher accuracy and the authors walk through different steps taken to ensure data quality. Overall this study is highly useful to the community and will be a guide to others collecting observations and comparing with satellite retrievals in remote locations. I recommend some minor revisions primarily to increase details and improve clarity.

[Figure]

General comments

G1 – The word "calibrate" has been used throughout to describe reducing the mismatch between TCCON and EM27/SUN retrievals, when in most cases "scale" would be a better choice. Calibration is generally reserved for something directly observed of known accuracy, and XCO2 is neither. In some situations "calibration" is okay so the wording is not too awkward (e.g., calibration curve), but should be changed in most instances.

G2 – It seems that differences in averaging kernels (AKs) and a priori profiles have not been considered in this study. E.g., on P2L35 it is stated measurements can be compared directly. It should be explicitly stated why AKs were not considered. Perhaps the a priori profiles are close enough to the true profiles *in this location* that accounting for these different sensitivities would make little difference? Also, it seems observations were first averaged rather than directly compared from both the EM27/SUN and GOSAT? It would also be interesting to know how much of the H to M gain bias is from AKs, if any.

Specific comments

S1, P1L7: State length of campaign here

S2, P1L~1: How exactly are the different gains chosen and used for GOSAT? Is gain chosen in real time by an onboard sensor, or are all gains recorded and the retrieval just picks the best one later? How/why did the gains change in Fig. 1 going from v 2.60 to v 2.72?

S3, P2L16: What does this mean? There are still some large cities in deserts.

S4, P2L20: This goes back to the question of how gain is chosen, but are M-gain regions always exclusively semi-arid? If so where does the classification of climate come from in the algorithm?

S5, P2L20: Carbon cycle studies could be (and have been) carried out with biased

satellite retrievals, even if the bias is "small." I think the focus here though is if the goal is to continuously improve the accuracy of these studies, then accuracy of satellite retrievals needs to be improved as well. What would happen if all observations over high albedo were not available? Likely results would be biased, and would lead to misinterpretation.

S6, P2L25: Why is high reflectivity a challenge for satellite observations? How does the extra reflectance influence the retrievals negatively? (Same question on P15L12)

S7, P2L25: How does having challenging observations naturally lead to their improvement? (Maybe the meaning here is it rather an opportunity?)

S8, P2L27: Why are such studies needed?

S9, P3L1-2: Suggest you pick a notation for EM27 or EM27/SUN early on and stay consistent throughout the entire paper.

S10, P3L2: Are v 2.60 data bias corrected? How?

S11, P3L1-2: Suggest you move this sentence to later on in the paragraph where again it states EM27/SUN retrievals were compared with GOSAT.

S12, P3L9: How was the EM27/SUN retrofitted?

S13, P3L10: suggest you include 2 decimals on latitude. Also, include dates here.

S14, P4L4: Specify these are column measurements.

S15, P4L7: Was the 2nd detector present for this study? Give full spectral range for this detector.

S16, P4L7-8: Suggest you move this sentence before the previous one. Currently it sounds like O2 is measured on the secondary detector.

S17, P4L9: What is the spectral resolution of TCCON measurements?

S18, P5L2: "greenhouse gas (CO2 and CH4) total columns" could simply be replaced

with "XCO2 and XCH4."

S19, P5L~23: Suggest you describe how Xair can provide info on stability (e.g., as a measure of retrieved O2, which is not particularly variable in dry air).

S20, P6L2: Clarify "specific" here. Is it only soundings within 100 km of the EM27? Are they compared individually or averaged together?

S21, P7L4: Frey et al., 2018 (https://doi.org/10.5194/amt-2018-146) would also be an appropriate reference here.

S22, P8: Was there any other alignment of the EM27 or TCCON instrument during this period?

S23, P10Fig6: Specify what points represent in caption. Daily averages? Daily averages within certain sza?

S24, P11L2: How were GOSAT data interpolated? (spatially? temporally? method?)

S25, P11L3: This is an unweighted mean?

S26, P11L12: Quantify approximate magnitude of annual increase here.

S27, P12L1: Did the number of H-gain observations decrease, or just the number of successful retrievals? If it's just the retrievals could it be an increased failure of convergence?

S28, P12L2: Apr-Aug mean in absolute number of soundings would also be useful here, same on line 4.

S29, P12L1-7: Seems a better sentence order could be: less rain -> less vegetation -> bright surface/more M-gain (rather than less vegetation -> bright surface/more M-gain -> less rain).

S30, P12Table1: What do the seconds mean on the measurement times? Start time? Central time? Seems the measurements could take up to 20 seconds. Also, the purpose of the first and second to last rows is not clear if no GOSAT soundings were acquired.

S31, P13Fig9: I do not really like lines fit through single points. Granted the intercept is forced through zero, but I think this information would be better for a table.

S32, P14L2: Where did these values come from?

S33, P14L7: satellite measurements -> satellite and ground-based measurements

S34, P15L2: What about Wollongong? Darwin? Reunion? Showa? Arrival Heights? Ascension? (Fig. 1: https://doi.org/10.5194/amt-9-2381-2016)

S35, P15L4-5: This sentence seems a bit redundant with the first sentence.

S36, P16L~1: Also add a description on how GOSAT data can be acquired

Technical

T1, P1L7: values, a -> values, another

T2, P1L11: improve -> better understand (or estimated -> estimation of)

T3, P2L4: along with -> and there has been

T4, P2L6: version 2.72 -> version 2.72 Xgas retrieval algorithm (w/o sentence seems to be missing a subject)

T5, P2L8: precise -> precise and accurate (?)

T6, P2L17: by anthropogenic -> by recent anthropogenic

T7, P2L20: are -> is

T8, P2L25: provides -> has

T9, P2L25: which challenges -> which is a challenge for

T10, P2L27: could be -> is

T11, P2L28: portable -> portable solar-viewing

T12, P3L8: "in the urban area" seems redundant, maybe omit?

T13, P3L17: average high -> average daily high

T14, P3L18-19: omit parenthetical comment (already on page 2, and Fig 1 caption)

T15, P3L21: maybe omit "reasonably accessible" as this is somewhat vague

T16, P5L19-20: suggest "s" on Ps and "a" on Na be subscripts

T17, P5L21: can -> is (?)

T18, P6L6: retrievals -> retrievals separately.

T19, P8L6: should give -> gives

T20, P11L5: were -> are

T21, P12L1: H-gain -> the number of H-gain (or better, -> there were 23% fewer H-gain)

T22, P12L2: Could omit "derived"

T23, P14L14: Could omit "in Alice Springs"

T24, P14L15: cover -> enclosure (x2)

T25, P14L17: omit "the assumption"

T26, P15L8: suggest omission of "and on other future satellites"

T27, P15L11: provides -> has

T28, P15L12: to -> for

T29, P15L12: this data is especially needed -> observations are especially needed here

T30, P16L7: calculated -> characterized

T31, P16L9: co-funded -> co-acquired funding for (?)

T32, P16L14: define RA

T33, P16L14: advise -> advice

Other notes/optional

O1, P2L15: The population of the greater LA area includes parts of other counties. https://www.citypopulation.de/world/Agglomerations.html lists the population as 17.7 million.

O2, P5L3: The authors may also consider publishing a description of the design and/or control software in the future (e.g., compare https://doi.org/10.5194/amt-11-2173-2018). Such a project/paper could be especially useful to the community if the control software were open source, and fully automated neither of which have been done yet. Example automation of OPUS: https://doi.org/10.1364/AO.57.000689, example of alternate solar tracking software: http://hdl.handle.net/10222/64642, Chapter 4.

O3, P6Fig2: More details on this schematic could be useful, such as the path of light, and parts that move/rotate or disassemble (it looks like a seam at the "v" part?) An actual picture at Alice Springs could be nice if available. A picture would provide the readers an idea if any precautions needed to be taken to prevent interference from unique Australian fauna, such as fencing or placement away from trees.

O4, P9L9-11: The ME at MOPD values seem particularly small, compared to typical values around 98-99% (Frey at al, 2018). It may be beneficial to realign the spectrometer. Though changes in Xair do not look large. . .

O5, P10: Future measurements in Alice Springs may be useful to help derive an airmass correction for all EM27/SUN instruments. This dataset may not be ideal though

since the ME at MOPD seems to differ from most other EM27/SUN instruments, and is only for part of one season.

O6, P14L2: A histogram of the standard deviations for the different gains could be useful.

---

## Referee Comment (RC2) · Anonymous Referee #2 · 11 Feb 2019

**General comments** The manuscript "Satellite and Ground-based Measurements of XCO2 in a Remote Semi-Arid Region of Australia" by Velazco et. al. describes the campaign-based deployment of an EM27/SUN near infrared spectrometer to gather targeted validation data for the Greenhouse Gases Observation Satellite (GOSAT). The paper also describes calibration of the portable spectrometer against a Total Carbon Column Observing Network (TCCON) instrument to demonstrate traceability to WMO trace gas scales and goes on to provide guidance on the ideal length of such a field campaign needed to achieve a required uncertainty in the satellite bias estimate.

The manuscript is laid out in a logical fashion and is well written.

[Figure]

Overall the manuscript documents a very useful piece of work that has been designed to answer a pertinent science question, and the resulting dataset, and I would recommend its publication subject to some minor modifications and clarifications.

**Specific comments**

It appears that the Authors have not smoothed the XCO2 retrievals from the various instruments to account for differences in the averaging kernels and priors (if these differ between the retrieval schemes) per the method of Rogers and Connor (2003) as implemented by Wunch et. al. (2010). It may be that the effect of this process would be negligible, however this should still be discussed and quantified.

Introduction and Fig. 1. Is there a reason why there appear to be more v2.72 M-gain retrievals?

There should be some commentary on how the GOSAT specific point observations differ from normal GOSAT observations in Section 3.3.

In section 4.1.1 comparing the retrievals of Xair between the EM27 and TCCON instruments, it is noted that there are both airmass and seasonal variations in Xair. It seems likely that at least some of the seasonal dependence might be caused by the differing ranges of solar zenith angles that are observed throughout the year. It would be interesting to plot Fig 5 for a limited range of solar zenith angles to identify if the airmass dependence is the only reason for the seasonal variation. In any case, it would be useful to have a few more details about the year of inter-comparison measurements in the introductory paragraph of Sect. 4.1 e.g. number of days of measurements, total number of measurements from each instrument and whether they were operated for the same periods on each day.

Section 4.1.2 and Fig. 6, what time averaging is applied to the data in Fig. 6? The colour scale and the size of the $1\sigma$ uncertainties suggest daily averages, but this is not made explicit. There also appears to be quite a large variety in the magnitude of the

$1\sigma$ values in Fig. 6. which would bear explanation.

Several times during the manuscript calibration factors are presented in the form $INST1 = F * INST2$ it would be useful to have some indication of the uncertainty of the F value, or the goodness of the regression fit used to derive it. Similarly, for the drift mentioned at P9L17.

Section 4.4. describing the length of campaign required to reduce the error in the bias estimates is interesting but would benefit from some discussion of what the target for accuracy and precision in the bias estimate should be.

Conclusions: this section should have some comment relating to the presented dataset specifically. Also, as an interested reader it would be good to know if there are any plans to repeat the field campaign and build the dataset.

P15L3, the sentence starting "With the exception of Lauder..." should be qualified.

Data availability: what about the GOSAT data?

**Technical corrections**

P2L24 "atmospheric conditions that can serve as **a** calibration point...

P4 Fig. 1 caption: Start with "Location of.." to indicate that these do not represent the retrieved values, for consistency, use XCO2 and XCH4.

P4L7 "below 5000 cm$^{-1}$ **which** allows for..."

P5L20 The last sentence on this page is disjointed and difficult to read, consider revising.

P6L4 "However, to construct **the** time-series..."

P6L4 should refer to Fig. 8, not Fig. 7.

P8 Fig. 4 caption: it should be sufficient to say month number and delete "in a year".

P9L12 Sentence can be finished after "campaign" to avoid repetition.

P10L7 "...retrievals results **in** lower..."

P11 Fig. 7 caption: use XCO2 for consistency.

P11 Fig. 7 caption: "days on the x-axis **are** not"

P11 Fig. 7 caption: pluralise interruption

P12L2 Start the sentence with "The number of..." to clarify that the retrieved values are not reducing.

P16L14 Explain what RA means.

P18L4 correct the rendering of the subscript

P18L33 The DOI is repeated

P19L6 insert a space between inverse and models

―――――――――――――――

---

## Author Response (AR1)

This is a single pdf that includes point-by-point responses to the reviews and a marked-up manuscript version showing the changes to the original manuscript.

Below we have included the full text of the review (black text), interspersed with our responses addressing the reviewer's specific comments and changes to the manuscript in italicised blue font.

**Reviewer 1 Comments:**

G1 – The word "calibrate" has been used throughout to describe reducing the mismatch between TCCON and EM27/SUN retrievals, when in most cases "scale" would be a better choice. Calibration is generally reserved for something directly observed of known accuracy, and XCO2 is neither. In some situations "calibration" is okay so the wording is not too awkward (e.g., calibration curve), but should be changed in most instances.

We agree and have made the changes, for example in Section 4.1.2. we made changes to a few sentences:

"From the EM27 and TCCON measurements in 2016, we derived a calibration scaling factor for the EM27. We calibrated scaled the EM27 measurements to Wollongong TCCON data because the Wollongong XCO2 has been calibrated against aircraft profile measurements that are traceable to the WMO in-situ scale (Wunch et al., 2010)."

"From this exercise, we arrive at a *calibration* scaling factor of EM27=0.9954\*TCCON."

"Therefore, we are confident that the *calibration* scaling factor relative to TCCON was consistent during the campaign with that derived relative to Wollongong TCCON."

Also under Section 4.3

*"From here, we derived a calibration scaling factor of 0.9927\*EM27 for GOSAT H-gain and 0.9983\*EM27 for the GOSAT M-gain,..."*

G2 – It seems that differences in averaging kernels (AKs) and a priori profiles have not been considered in this study. E.g., on P2L35 it is stated measurements can be compared directly. It should be explicitly stated why AKs were not considered. Perhaps the a priori profiles are close enough to the true profiles \*in this location\* that accounting for these different sensitivities would make little difference? Also, it seems observations were first averaged rather than directly compared from both the EM27/SUN and GOSAT? It would also be interesting to know how much of the H to M gain bias is from AKs, if any.

We've added to Section 4.1.2, 2nd paragraph:

"We did not consider the averaging kernels in this work. The averaging kernels of the EM27 have been previously presented and compared to TCCON in a study by Hedelius et al. (2016). In their study, they found that although there are differences in the TCCON and EM27 averaging kernels, the effect of the differences in averaging kernels from the top of the atmosphere cancel out the effect of differences at the bottom."

We have also reworded the last sentence in Section 4th paragraph:

"The Alice Springs measurements are unique considering that they have been collected from a clean desert environment and can be directly compared to GOSAT retrievals using the M-gain measurements."

То

The Alice Springs measurements are unique considering that they have been collected from a clean desert environment where GOSAT M-gain soundings are also abundant and close enough (<100 km) to compare with the EM27.

**Specific comments**

S1, P1L7: State length of campaign here *Done, thanks*

S2, P1L\_1: How exactly are the different gains chosen and used for GOSAT? Is gain chosen in real time by an onboard sensor, or are all gains recorded and the retrieval just picks the best one later?

**We have revised the last sentences of the first paragraph in the Introduction:**

"These gain settings are prespecified at certain locations because GOSAT does not observe both M and H gains simultaneously. For the majority of the soundings over land, H-gain is used. GOSAT Mgain retrievals over land are used over surfaces that are bright in the SWIR such as deserts and semiarid regions. However, a bias between GOSAT M- and H-gain retrievals of XCO2 has been reported, along with a lack of M-gain validation with TCCON (Yoshida et al., 2013). In Australia where surface reflectivity values are generally high, GOSAT was configured to observe using M gain for much of the land surface, at first to avoid detector saturation. However, it was found that some observations using H gain did not result in saturated signals and were still useful. Therefore from 2012-Feb-11, GOSAT started to perform alternate observations using both H and M gains in order to investigate the differences from retrievals between these two gain settings. Therefore from 2012-Feb-11, GOSAT started to perform alternate observations using both H and M gains in order to investigate the differences from retrievals between these two gain settings. Therefore from 2012-Feb-11, GOSAT

How/why did the gains change in Fig. 1 going from v 2.60 to v 2.72?

This is a good question, thanks. So under Section 2 "Alice Springs Australia Site Description", paragraph 1, we've added:

"Note that for Ver. 02.72 FTS SWIR L2 retrievals, upgraded input and reference products were used. For example, there was an improvement in the spatial resolution of the cloud flagging procedure, which employs the CAI L2 (Cloud and Aerosol Imager, Level 2) data. This improvement resulted in better screening of the data and may have resulted in an increased number of soundings that were passed for the Ver. 02.72 FTS SWIR L2 retrievals."

*These changes are described in the release notes:* https://data2.gosat.nies.go.jp/doc/documents/ReleaseNote\_FTSSWIRL2\_V02.72\_en.pdf

S3, P2L16: What does this mean? There are still some large cities in deserts.

*True, this sentence was poorly constructed, we now changed it to: "The world's deserts and semi-arid regions encompass large areas that are mostly undisturbed by anthropogenic emissions and are important for understanding the carbon cycle."*

S4, P2L20: This goes back to the question of how gain is chosen, but are M-gain regions always exclusively semi-arid? If so where does the classification of climate come from in the algorithm?

Thanks for the question. We believe we've addressed this above

S5, P2L20: Carbon cycle studies could be (and have been) carried out with biased satellite retrievals, even if the bias is "small." I think the focus here though is if the goal is to continuously improve the accuracy of these studies, then accuracy of satellite retrievals needs to be improved as well. What would happen if all observations over high albedo were not available? Likely results would be biased, and would lead to misinterpretation.

*This is correct! Thank you for this very important point. We would like to use it in the manuscript so we added:*

"Recently, the importance of semi-arid regions in the carbon cycle inter-annual variability has been highlighted (Poulter et al., 2014). If the goal is to continuously improve the accuracy of carbon cycle studies, then the accuracy of satellite retrievals needs to be improved as well because if observations over high albedo were not available or biased, flux estimates would likely be biased as well and would lead to misinterpretation. Therefore, measurements over M-gain regions are needed by the satellite community (Yoshida et al., 2013) and highlights the significance of this study.

S6, P2L25: Why is high reflectivity a challenge for satellite observations? How does the extra reflectance influence the retrievals negatively? (Same question on P15L12) S7, P2L25: How does having challenging observations naturally lead to their improvement? (Maybe the meaning here is it rather an opportunity?)

Thanks for these good questions we would like to address them at the same time. In the manuscript, we explained this in the text as below:

"The desert environment provides high surface reflectivity, which challenges satellite retrievals because aerosols, depending on type, can lead to an effect called optical path lengthening and this effect is dominant in regions with high albedo (Yoshida et al., 2013). Recently, Iwasaki et al. (2019) showed that there is an increase in the  $XCO_2$  retrievals using their PPDF-S algorithm when the albedo at 1.6µm was high, implying that the retrieved  $XCO_2$  is strongly related to the surface albedo. This challenge leads to the improvement of satellite retrievals."

S8, P2L27: Why are such studies needed?

Thanks, we have expanded this sentence to:

"Benchmark measurements and pilot studies for desert sites are needed to assess the benefit and feasibility of such sites because setting up a TCCON site in remote deserts will be difficult logistically and financially."

S9, P3L1-2: Suggest you pick a notation for EM27 or EM27/SUN early on and stay consistent throughout the entire paper.

Good point, we have done this early on in the Introduction by writing:

"We address this need by utilizing a well-established portable spectrometer, an EM27/SUN by Bruker Optics GmbH (Gisi et al., 2012; Frey et al., 2015; Hase et al., 2015; Hedelius et al., 2016; Frey et al., 2018), which was retrofitted with a protective fairing and automated solar tracker hatch for operations in a harsh environment. For brevity, we will refer to this instrument as EM27."

S10, P3L2: Are v 2.60 data bias corrected? How?

Actually, both are NOT bias corrected, so we rewrote that part: "We compare NIES GOSAT retrieval versions 2.60 and the new version 2.72, which are both not bias corrected."

S11, P3L1-2: Suggest you move this sentence to later on in the paragraph where again it states EM27/SUN retrievals were compared with GOSAT.

Thanks, we moved this after mentioning Section 4.

S12, P3L9: How was the EM27/SUN retrofitted?

*We added this in the introduction:*

"...retrofitted with a protective fairing and automated solar tracker hatch for operations in a harsh environment."

S13, P3L10: suggest you include 2 decimals on latitude. Also, include dates here.

Done, thanks.

S14, P4L4: Specify these are column measurements.

Well spotted, thanks.

S15, P4L7: Was the 2nd detector present for this study? Give full spectral range for this detector.

No, it wasn't present so we deleted that sentence, which takes care of the next issue

S16, P4L7-8: Suggest you move this sentence before the previous one. Currently it sounds like O2 is measured on the secondary detector.

True indeed, thanks, we did this.

S17, P4L9: What is the spectral resolution of TCCON measurements?

*TCCON* uses a maximum optical path difference (MOPD) of 45 cm, corresponding to a spectral resolution of  $0.02 \text{ cm}^{-1}$ . We've added it now, thanks.

S18, P5L2: "greenhouse gas (CO2 and CH4) total columns" could simply be replaced with "XCO2 and XCH4."

**Thanks, done.**

S19, P5L\_23: Suggest you describe how Xair can provide info on stability (e.g., as a measure of retrieved O2, which is not particularly variable in dry air).

Thank you, this is a very good and important point. We improved the Xair description according to this suggestion.

"Xair is a good indicator of instrument stability and changes in spectrometer alignment because 1) VC\_air is calculated using the surface pressure, which is independently measured by a pressure sensor to better than 0.3 hPa, keeping accuracy over long periods (Wunch et al., 2011); and 2) the atmospheric oxygen column is not particularly variable in dry air, hence the retrieved VC\_O2 by the spectrometer should be close to constant. From Eq. 3, it follows that a perfectly accurate measurement would lead to an Xair value of unity, however, due to spectroscopic limitations, the actual value is approximately 0.98 for all TCCON sites.

S20, P6L2: Clarify "specific" here. Is it only soundings within 100 km of the EM27? Are they compared individually or averaged together?

**We elaborated this paragraph to:**

"The TANSO-FTS on GOSAT has a two-axis pointing system. Normally, TANSO-FTS follows an M shaped grid on a 5-point cross-track scan mode. By using this pointing system to vary the observation geometry, it is able to observe specific points, i.e. it can view targets with angles up to  $\pm 20^{\circ}$  along the satellite track and by  $\pm 30^{\circ}$  across the track. Specific point observations over Alice Springs were requested from July 2016, in preparation for the campaign in September. Five locations within 100 km of the center of Alice Springs were targeted (see Fig. 3).

S21, P7L4: Frey et al., 2018 (https://doi.org/10.5194/amt-2018-146) would also be an appropriate reference here.

Thanks, done.

S22, P8: Was there any other alignment of the EM27 or TCCON instrument during this period?

Good question. In the revised version, we state:

"We note that there were no re-alignments done on the EM27 and TCCON instruments during this period, however the EM27 clamshell cover and fairing were fitted on the EM27 in February 2016, it is possible that this may have affected the alignment resulting in a shift in the Xair."

S23, P10 Fig6: Specify what points represent in caption. Daily averages? Daily averages within certain sza?

In the Figure caption, we now write:

"Weighted hourly averaged  $XCO_2$  from TCCON vs  $XCO_2$  from EM27 in Wollongong for 2015-2016. To avoid noisy data, only measurements corresponding to TCCON with retrieved Xair values within 0.9783 and 0.9853 were selected (Xair values within the 2nd to 98th percentile)"

S24, P11L2: How were GOSAT data interpolated? (spatially? temporally? method?)

We changed this sentence. The word "interpolated" was removed because it can be confusing. Instead we write:

"To demonstrate the variability of  $XCO_2$  in the region, we plot in Fig. 8 the daily mean time series of GOSAT soundings using these two coincidence criteria: 1) all soundings within a 1000-km radius centered at the BOM facility in Alice Springs and 2) all soundings measured within the same day (local time). Each data point considered in the calculation of the daily mean is weighted by the corresponding reported retrieval error for that particular GOSAT sounding."

S25, P11L3: This is an unweighted mean?

No, as mentioned above, each data point considered in the calculation of the daily mean is weighted by the corresponding retrieval error for each GOSAT sounding falling within the spatial and temporal coincidence criteria. This associated error is reported in the GOSAT data.

S26, P11L12: Quantify approximate magnitude of annual increase here.

*We have quantified from the slope of line in Fig. 8 the following XCO*2 *annual increase: H-gain: 2.2815 ppm/year; intercept: 132.9996 M-Gain: 2.2829 ppm/year; intercept: 134.9174*

So, at the end of the first paragraph in Sec. 4.3, we wrote:

"We confirm that the M-gain retrievals are biased high (around 2 ppm) compared to the H-gain retrievals. Nevertheless, both least squares fitted lines to the H and M-gain retrievals show an increase of about 2.28 ppm/year with y-intercept values at 133.0 ppm (H-gain) and 134.92 ppm (M-gain)."

S27, P12L1: Did the number of H-gain observations decrease, or just the number of successful retrievals? If it's just the retrievals could it be an increased failure of convergence?

The number of successful retrievals increased. At the end of Sec 2, we added:

"there was an improvement in the spatial resolution of the cloud flagging procedure, which employs the CAI L2 (Cloud and Aerosol Imager, Level 2) data. This improvement resulted in better screening of the data and may have resulted in an increased number of soundings that were passed for the Ver. 02.72 FTS SWIR L2 retrievals."

S28, P12L2: Apr-Aug mean in absolute number of soundings would also be useful here, same on line 4.

We agree and included this (please refer to the reply below).

S29, P12L1-7: Seems a better sentence order could be: less rain -> less vegetation -> bright surface/more M-gain (rather than less vegetation -> bright surface/more M-gain -> less rain).

**Thank you, we agree. This paragraph has now been changed to:**

Rainfall records indicate that April-August 2011 had the least amount of rainfall on record in Alice Springs after 2002, which was only broken by 2015, then 2017 (http://www.bom.gov.au/climate/data/, station number 015590). This significant absence of rain in the region could result in stunted vegetation growth or mortality, which could have led to brighter surfaces. Bright surfaces mean more M-gain measurements and this may explain the 74% increase in M-gain soundings in April-August 2011 compared to the April-August 2010-2017 average of c. 220 soundings/month. In contrast, the number of H-gain retrievals seems to have diminished around 2011, coinciding with the dry months starting from April 2011. The number of H-gain soundings from April-August 2011 was 23% fewer compared to the April-August average from the years 2010-2017 (c. 450 soundings/month).

S30, P12Table1: What do the seconds mean on the measurement times? Start time? Central time? Seems the measurements could take up to 20 seconds. Also, the purpose of the first and second to last rows is not clear if no GOSAT soundings were acquired.

**We removed the seconds and simply wrote:**

The times correspond to the times when the satellite is directly above the site. The satellite normally performs 5 observations points across track, with an interferometric scan time lasting 4 seconds (Shiomi et al. 2006). But specific point observations deviate from this pattern by pointing and maximizing observations near the target.

S31, P13Fig9: I do not really like lines fit through single points. Granted the intercept is forced through zero, but I think this information would be better for a table.

Yes, the intercept is forced through zero because both instruments are expected to produce zero  $XCO_2$  when there is no CO2 absorption. We added the sentence to Sec. 4.1.2:

"The retrieval method is predicted to be both linear and have zero intercept (Wunch et al., 2010). Therefore, we fit hourly mean data from TCCON and EM27 employing linear least squares and force a zero intercept."

We would like to keep the figure because for future campaigns and comparisons with more data points we would like to use this as reference.

S32, P14L2: Where did these values come from?

*We clarified this item in the paper and also considered the standard errors of the weighted mean for version 2.72:*

"The standard errors of the weighted hourly mean GOSAT specific point observations are: 0.25 ppm and 0.26 ppm for M-gains (versions 2.6 and 2.72, respectively) and 0.29 ppm and 0.28 ppm for H-Gains (versions 2.6 and 2.72, respectively). These values are taken from the averaged standard errors of the weighted hourly means from all specific point observation data falling within 100 km from the site for 2 Sep. 2015 - Feb. 2017".

S33, P14L7: satellite measurements -> satellite and ground-based measurements

Done, thanks.

S34, P15L2: What about Wollongong? Darwin? Reunion? Showa? Arrival Heights? Ascension? (Fig. 1: https://doi.org/10.5194/amt-9-2381-2016)

**We elaborated this sentence to:**

"With the exception of Lauder, New Zealand, there are no permanently dedicated ground targets for satellite calibration and validation in the Southern Hemisphere that is far enough from water (to avoid a possible land-ocean bias in retrievals) and surrounded by homogeneous topography (to avoid a possible altitude bias)."

S35, P15L4-5: This sentence seems a bit redundant with the first sentence. *Agreed, we removed this sentence*

S36, P16L\_1: Also add a description on how GOSAT data can be acquired *Done. thanks*

**Technical**

T1, P1L7: values, a -> values, another *done, thanks*.

T2, P1L11: improve -> better understand (or estimated -> estimation of) *thanks, changed to "better understand"*

T3, P2L4: along with -> and there has been *agreed, thanks*.

T4, P2L6: version 2.72 -> version 2.72 Xgas retrieval algorithm (w/o sentence seems to be missing a subject) *Thanks, sentence modified.*

T5, P2L8: precise -> precise and accurate (?) *Yes, that is right, change done.*

T6, P2L17: by anthropogenic -> by recent anthropogenic *Thanks, this is better.*

T7, P2L20: are -> is *Done*.

T8, P2L25: provides -> has *Done, thanks*.

T9, P2L25: which challenges -> which is a challenge for *Corrected, thank you.*

T10, P2L27: could be -> is *Changed*.

T11, P2L28: portable -> portable solar-viewing *Done, thanks.*

T12, P3L8: "in the urban area" seems redundant, maybe omit? *Agreed, done.*

T13, P3L17: average high -> average daily high *Done, thanks*

T14, P3L18-19: omit parenthetical comment (already on page 2, and Fig 1 caption) *We agree and omission done*.

T15, P3L21: maybe omit "reasonably accessible" as this is somewhat vague *We agree and did this.*

T16, P5L19-20: suggest "s" on Ps and "a" on Na be subscripts *Changed, thanks.*

T17, P5L21: can -> is (?) *Yes, thanks. The sentence has been changed.*

T18, P6L6: retrievals -> retrievals separately. *This is better, thanks*

T19, P8L6: should give -> gives *Done, thanks.*

T20, P11L5: were -> are *Done, thanks.*

T21, P12L1: H-gain -> the number of H-gain (or better, -> there were 23% fewer Hgain) T22, P12L2: Could omit "derived"

Thanks, to accommodate both of these, we changed the sentences to: "In contrast, the number of H-gain retrievals seems to have diminished around 2011, coinciding with the dry months starting from April 2011. H-gain soundings from April-August 2011 were 23% fewer compared to the April-August mean from the years 2010-2017."

T23, P14L14: Could omit "in Alice Springs" *Thanks, done.*

T24, P14L15: cover -> enclosure (x2) *Done, thanks.*

T25, P14L17: omit "the assumption" *Omitted, thanks*

T26, P15L8: suggest omission of "and on other future satellites" *Suggestion accepted, thanks.*

T27, P15L11: provides -> has *Done*.

T28, P15L12: to -> for *Done, thanks*

T29, P15L12: this data is especially needed -> observations are especially needed here *This is an improvement, thanks*

T30, P16L7: calculated -> characterized *Done, thanks.*

T31, P16L9: co-funded -> co-acquired funding for (?) *Agreed, thanks*

T32, P16L14: define RA *Defined, thanks*

T33, P16L14: advise -> advice *Corrected, thanks.*

**Other notes/optional**

O1, P2L15: The population of the greater LA area includes parts of other counties. https://www.citypopulation.de/world/Agglomerations.html lists the population as 17.7 million.

Thank you very much for this information. The agglomerated population count represents the region better, so we changed it according to this.

O2, P5L3: The authors may also consider publishing a description of the design and/or control software in the future (e.g., compare https://doi.org/10.5194/amt-11-2173-2018). Such a project/paper could be especially useful to the community if the control software were open source, and fully automated neither of which have been done yet. Example automation of OPUS: https://doi.org/10.1364/AO.57.000689, example of alternate solar tracking software: http://hdl.handle.net/10222/64642, Chapter 4.

Yes indeed, the EM27 enclosure has undergone several revisions, our aim is to have 3D drawings that we can share to the community. The Davis weather station has a built-in programmable logic; we are also planning to distribute the configuration.

O3, P6Fig2: More details on this schematic could be useful, such as the path of light, and parts that move/rotate or disassemble (it looks like a seam at the "v" part?) An actual picture at Alice Springs could be nice if available. A picture would provide the readers an idea if any precautions needed to be taken to prevent interference from unique Australian fauna, such as fencing or placement away from trees.

Yes, there is a seam at the "V" part for easy access to the solar beam inlet. However, the design has changed a little bit, so we are currently working on the engineering drawings that we can distribute. We have added a picture of the EM27 at the Alice Springs site as an inset to Fig 3. Fortunately, this site is just beside the airport and is fully fenced, so the noise and the fencing keep away large Australian fauna (e.g. Dingoes, Kangaroos, Sand goannas, etc.).

O4, P9L9-11: The ME at MOPD values seem particularly small, compared to typical values around 98-99% (Frey at al, 2018). It may be beneficial to realign the spectrometer. Though changes in Xair do not look large. . .

**Yes, well noted. Unfortunately, unlike the 125HR, alignment of the EM27 is still not well documented in the community, therefore for a re-alignment we have to send the EM27 to Karslruhe Institute of Technology (to Frey et al.) or directly to Bruker Germany. This was done in 2017.**

O5, P10: Future measurements in Alice Springs may be useful to help derive an airmass correction for all EM27/SUN instruments. This dataset may not be ideal though since the ME at MOPD seems to differ from most other EM27/SUN instruments, and is only for part of one season.

Deriving an airmass correction is an excellent point. For future campaigns, we will make sure to have the EM27 re-aligned at Bruker or at Karlsruhe. A loan from the COCCON network (or others) may be a possibility as well.

O6, P14L2: A histogram of the standard deviations for the different gains could be useful.

This is a good idea. We followed this and added an inset to Fig 10. Also, we've added on Section 4.4,  $2^{nd}$  paragraph, sentences 1 to 3:

"Fig. 10 shows a plot of how the statistical errors in the estimated bias improve with the number of weeks in the campaign. The inset shows the normalized histogram of the reported GOSAT single sounding errors within 100 km of the site. Note that although H-gain retrievals have smaller errors on average, the amount of M-gain retrievals improves the standard error of the weighted mean."

**Reviewer 2 Specific comments**

It appears that the Authors have not smoothed the  $XCO_2$  retrievals from the various instruments to account for differences in the averaging kernels and priors (if these differ between the retrieval schemes) per the method of Rogers and Connor (2003) as implemented by Wunch et. al. (2010). It may be that the effect of this process would be negligible, however this should still be discussed and quantified.

**Thank you for this important comment. Under Section 4.1, we've added:**

"We did not consider the averaging kernels in this work. The averaging kernels of the EM27 have been previously presented and compared to TCCON in a study by Hedelius et al. (2016). In their study, they found that although there are differences in the TCCON and EM27 averaging kernels, the effect of the differences in averaging kernels from the top of the atmosphere cancel out the effect of differences at the bottom. Further work and more measurements may be necessary to better understand the cause of this phenomenon but this is beyond the scope of this study."

Introduction and Fig. 1. Is there a reason why there appear to be more v2.72 M-Gain retrievals?

Yes, this is a good question, thanks. So under Section 2 "Alice Springs Australia Site Description", paragraph 1, we've added:

"Note that for Ver. 02.72 FTS SWIR L2 retrievals, upgraded input and reference products were used. For example, there was an improvement in the spatial resolution of the cloud flagging procedure, which employs the CAI L2 (Cloud and Aerosol Imager, Level 2) data. This improvement resulted in better screening of the data and may have resulted in an increased number of soundings that were passed for the Ver. 02.72 FTS SWIR L2 retrievals."

These changes are described in the release notes: https://data2.gosat.nies.go.jp/doc/documents/ReleaseNote FTSSWIRL2 V02.72 en.pdf

There should be some commentary on how the GOSAT specific point observations differ from normal GOSAT observations in Section 3.3.

**Good point, therefore under Section 3.3, we wrote:**

"The TANSO-FTS on GOSAT has a two-axis pointing system. Normally, TANSO-FTS follows an M shaped grid on a 5-point cross-track scan mode. By using this pointing system to vary the observation geometry, it is able to observe specific points, i.e. it can view targets with angles up to  $\pm 20^{\circ}$  along the satellite track and by  $\pm 30^{\circ}$  across the track. Specific point observations over Alice Springs were requested from July 2016, in preparation for the campaign in September. Five locations within 100 km of the center of Alice Springs were targeted (see Fig. 3).

In section 4.1.1 comparing the retrievals of Xair between the EM27 and TCCON instruments, it is noted that there are both airmass and seasonal variations in Xair. It seems likely that at least some of the seasonal dependence might be caused by the differing ranges of solar zenith angles that are observed throughout the year. It would be interesting to plot Fig 5 for a limited range of solar zenith angles to identify if the airmass dependence is the only reason for the seasonal variation.

Thanks, this is a good point. We revised Fig. 5 to show Xair values at SZA > 45°, SZA < 45° and all SZA and revised the section accordingly. The SZA dependence is small and within approximately 1%.

In any case, it would be useful to have a few more details about the year of inter-comparison measurements in the introductory paragraph of Sect. 4.1 e.g. number of days of measurements,

total number of measurements from each instrument and whether they were operated for the same periods on each day.

**Under Section 3.2 we added in the last paragraph:**

"In Wollongong, both the TCCON and EM27 solar tracker covers were opened and closed by the same pneumatic system. The same weather station provided the meteorological data that were used to pre-filter the data (e.g. fractional variation in solar intensity, wind speed and direction, pressure, etc). We did not filter the data according to solar zenith angles anymore, instead in addition to the prefilter, we filtered out noisy retrievals by selecting only those with Xair values within 0.5 and 1.5 because anything beyond that would be unrealistic in the atmosphere but most likely be the cause of an interference or obstruction."

And in Section 4.1, first paragraph, third sentence, we wrote:

"Here, we focus on comparisons of Xair and  $XCO_2$  from measurements spanning almost one year (Nov. 2015 to Sep. 2016)" under varied environmental conditions. Both instruments normally measure at the same time, apart from interruptions due to occasional mid-infrared measurements with the 125HR or rare software glitches (e.g. JAVA issues)."

Section 4.1.2 and Fig. 6, what time averaging is applied to the data in Fig. 6? The colour scale and the size of the 1 $\sigma$  uncertainties suggest daily averages, but this is not made explicit.

Actually, these are hourly averages, so in the text we wrote:

"Therefore, we fit hourly mean data from TCCON and EM27 employing linear least squares and force a zero intercept. The standard errors of the weighted means are used as weights in the fit. From this exercise, we arrive at a scaling factor of EM27=0.9954\*TCCON"

There also appears to be quite a large variety in the magnitude of the  $1\sigma$  values in Fig. 6. which would bear explanation. Several times during the manuscript calibration factors are presented in the form INST1 = F \* INST2 it would be useful to have some indication of the uncertainty of the F value, or the goodness of the regression fit used to derive it. Similarly, for the drift mentioned at P9L17.

We've added error bars on the inset plot for the drift and a Pearson R correlation coefficient (R=1) for the least squares fit in Fig. 6. The uncertainty in the F value arising from the linear fit, while forcing the intercept to zero, is very small (<1x10e-6), so we did not include it anymore.

Section 4.4. describing the length of campaign required to reduce the error in the bias estimates is interesting but would benefit from some discussion of what the target for accuracy and precision in the bias estimate should be.

*Thanks, this is a good point. We wrote 0.1 ppm as reference and we cite the work of Miller at al., 2007 and added on Sec 4.4, 2nd paragraph:*

"Miller (2007) showed that a comparison of surface  $CO_2$  concentration data and  $XCO_2$  data flux inversions clearly reveals a land-ocean bias in the  $XCO_2$  retrievals, even when the bias is only 0.1 ppm".

Conclusions: this section should have some comment relating to the presented dataset specifically. Also, as an interested reader it would be good to know if there are any plans to repeat the field campaign and build the dataset.

We included in the conclusion, 2nd to last sentence: "Subject to funding and interest from the community, a repeat of the campaign is possible." P15L3, the sentence starting "With the exception of Lauder..." should be qualified.

Thanks, we reworded this whole sentence to: "With the exception of Lauder, New Zealand, there are no permanently dedicated ground targets for satellite calibration and validation in the Southern Hemisphere that is far enough from water (to avoid a possible land-ocean bias in retrievals) and surrounded by homogeneous topography (to avoid a possible altitude bias)"

Data availability: what about the GOSAT data?

We included in the section "Data Availability" GOSAT data can be obtained from the GOSAT Data Archive Service (GDAS) after registration for access via: https://data2.gosat.nies.go.jp/index\_en.html

**Technical corrections**

P2L24 "atmospheric conditions that can serve as a calibration point... Yes, this is better, thanks.

P4 Fig. 1 caption: Start with "Location of.." to indicate that these do not represent the retrieved values, for consistency, use  $XCO_2$  and  $XCH_4$ . Good point, done. Thanks.

P4L7 "below 5000 cm-1 which allows for..."

We improved the sentences related to this to:

"For total column measurements of  $CO_2$ ,  $CH_4$ ,  $H_2O$  and  $O_2$  spectra in the near infrared, the instrument is fitted with an Indium Gallium Arsenide (InGaAs) detector dedicated to 5,000–12,000 cm–1. This also enables measurements of spectra covering the  $O_2$  bands necessary to derive column-averaged dry-air mole fractions of  $CO_2$  and  $CH_4$  similar to the method used by TCCON"

P5L20 The last sentence on this page is disjointed and difficult to read, consider revising. *Thanks, we agree and revised these sentences.*

P6L4 "However, to construct the time-series..." *Done, thanks*

P6L4 should refer to Fig. 8, not Fig. 7. *Well spotted, thanks.*

P8 Fig. 4 caption: it should be sufficient to say month number and delete "in a year". *Done, thanks*

P9L12 Sentence can be finished after "campaign" to avoid repetition. *Agreed, we've done this. Thanks.*

P10L7 "...retrievals results in lower..." Done, thank you.

P11 Fig. 7 caption: use XCO2 for consistency. *Done, thanks.*

P11 Fig. 7 caption: "days on the x-axis are not" *Corrected, thank you.*

P11 Fig. 7 caption: pluralise interruption *Corrected, thank you.*

P12L2 Start the sentence with "The number of..." to clarify that the retrieved values are not reducing. *Thanks for the suggestion, we corrected it.*

P16L14 Explain what RA means. *Done, thanks.*

P18L4 correct the rendering of the subscript We tried many times but there seems to be a difficulty in the subscripting between Overleaf and Mendeley. We will bring this up during the typesetting.

P18L33 The DOI is repeated *Corrected, thanks*

P19L6 insert a space between inverse and models *Done, thanks.*

**Satellite and Ground-based Measurements of $X_{\rm CO_2}$ in a Remote Semi-Arid Region of Australia**

Voltaire A. Velazco1, Nicholas M. Deutscher1, Isamu Morino2, Osamu Uchino2, Beata Bukosa1, Masataka Ajiro2, Akihide Kamei2, Nicholas B. Jones1, Clare Paton-Walsh1, and David W. T. Griffith1 1Centre for Atmospheric Chemistry, School of Earth, Atmospheric and Life Sciences, Faculty of Science, Medicine and Health, University of Wollongong, Australia 2Satellite Remote Sensing Section and Satellite Observation Center, Center for Global Environmental Research, National Institute for Environmental Studies (NIES), Onogawa 16-2, Tsukuba, Ibaraki 305-8506, Japan

Correspondence: Voltaire A. Velazco (voltaire@uow.edu.au)

[revised manuscript text omitted]

- 5 strongly related to the surface albedo. This challenge leads to the improvement of satellite retrievals. Despite the importance of desert locations like Central Australia in remote sensing, infrastructure support is not available and accessibility could be is a challenge. Benchmark measurements and pilot studies for desert sites are needed to assess the benefit and feasibility of such sites because setting up a TCCON site in remote deserts will be difficult logistically and financially. We address this need by utilizing a well-established portable solar-viewing spectrometer, an EM27/SUN by Bruker Optics GmbH
- 10 (Gisi et al., 2012; Frey et al., 2015; Hase et al., 2015; Hedelius et al., 2016; ?), which we retrofitted (Gisi et al., 2012; Frey et al., 2015; Hase, which was retrofitted with a protective fairing and automated solar tracker clamshell cover for operations in a harsh environment. For brevity, we will refer to this instrument as EM27. The instrument can measure spectra covering the spectral bands in the near infrared necessary to derive column-averaged dry-air mole fractions of CO2, CH4 and CO with sufficient stability for short-term campaigns (Hedelius et al., 2016). The instrument was transported to Alice Springs in Central Australia with
- 15 the primary objective of validating the GOSAT  $X_{CO_2}$  signal and making benchmark measurements in the region. The Alice Springs measurements are unique considering that they have been collected from a clean desert environment and can be directly compared to GOSAT retrievals using the where GOSAT M-gain measurements.

In this paper, we focus on measurements from the EM27 and GOSAT. We compare NIES GOSAT retrieval versions 2.60 and the new version 2.72, which are not bias corrected, with the retrievals from the soundings are also abundant and close enough (within 100 km) to compare with the EM27/SUN

20 (within 100 km) to compare with the EM27/SUN.

This manuscript is organized as follows. A description of Alice Springs and desert Australia is given in Section 2. In Section 3, we briefly discuss the instruments and methods, which are already well established. Results of measurement comparisons with the TCCON station in Wollongong and comparisons with the GOSAT M-gain and H-gain soundings in Alice Springs are shown and discussed under Section 4. We focus on  $X_{CO_2}$  measurements from the EM27 and GOSAT. We compare NIES

25 GOSAT retrieval versions 2.60 and the new version 2.72, which are both not bias corrected, with the retrievals from the EM27. Statistical calculations and recommendations for a future Alice Springs campaign are also under Section 4. We provide our conclusions in Section 5.

**2 Alice Springs Australia Site Description**

We conducted a measurement campaign with the retrofitted EM27 /SUN-system at the Australian Bureau of Meteorology
(BOM) facility in Alice Springs (23.723.79°S, 133.87133.89°E) near the end of September to the beginning of October from September 29 to October 6, 2016, with the primary objective of validating the GOSAT XCO2 signal in this semi-arid region. Alice Springs is located in central Central Australia, also called the "Red Centre". Next to Darwin and Palmerston, it is the third-largest town in the Northern Territory of Australia with a population of 23,726 in the urban area (2016 census). The

vegetation around Alice Springs is composed mostly of dry scrubby grassland. The Alice Springs terrain, consisting mostly of sandy plains with some areas of rocky highland, is bounded by several deserts; the Tanami desert to the north, Simpson desert to the east and southeast, the Great Victoria desert to the south and the Gibson desert to the west. In September, statistical data from the Australian Bureau of Meteorology (BOM) show an average monthly rainfall of 8.7 mm, average daily high and

- 5 low temperatures of 27.3 °C and 10.3 °C respectively and mean monthly sunshine of 300 hours. Collected GOSAT M-gain soundings over land for one whole year (black dots for version 2.6 and white squares for version 2.72) shown in Fig. 1 include a large part of this "Red Centre" including Alice Springs. Note that for Ver. 02.72 FTS SWIR L2 retrievals, upgraded input and reference products were used. For example, there was an improvement in the spatial resolution of the cloud flagging procedure, which employs the CAI L2 (Cloud and Aerosol Imager, Level 2) data. This improvement resulted in better screening of the data
- 10 and may have resulted in an increased number of soundings that were passed for the Ver. 02.72 FTS SWIR L2 retrievals. Apart from remote regions in North Africa, the Middle East and near-densely-populated areas in California USA, Central Australia is the only reasonably accessible region that provides a rich an abundant amount of M-gain soundings.